# Biological relevance of computationally predicted pathogenicity of noncoding variants

Li Liu[1], Maxwell D. Sanderford[2], Ravi Patel[2,3], Pramod Chandrashekar [1], Greg Gibson [4] & Sudhir Kumar[2,3]

Computational prediction of the phenotypic propensities of noncoding single nucleotide variants typically combines annotation of genomic, functional and evolutionary attributes into a single score. Here, we evaluate if the claimed excellent accuracies of these predictions translate into high rates of success in addressing questions important in biological research, such as fine mapping causal variants, distinguishing pathogenic allele(s) at a given position, and prioritizing variants for genetic risk assessment. A significant disconnect is found to exist between the statistical modelling and biological performance of predictive approaches. We discuss fundamental reasons underlying these deficiencies and suggest that future improvements of computational predictions need to address confounding of allelic, positional and regional effects as well as imbalance of the proportion of true positive variants in candidate lists.

[1] College of Health Solutions, Biodesign Institute, Arizona State University, Tempe, AZ, USA. [2] Institute for Genomics and Evolutionary Medicine, Temple University, Philadelphia, PA, USA. [3] Department of Biology, Temple University, Philadelphia, PA, USA. [4] School of Biological Sciences, Georgia Institute of Technology, Atlanta, GA, USA. These authors contributed equally: Li Liu, Maxwell D. Sanderford. These authors jointly supervised this work: Greg Gibson, Sudhir Kumar. Correspondence and requests for materials should be addressed to G.G. (email: greg.gibson@biology.gatech.edu) or to S.K. (email: s.kumar@temple.edu)

The past twenty years of sequencing effort has catalogued more than 300 million single nucleotide variants (SNVs) in the human genome, with many new and rare novel variants reported with each newly sequenced person[1]. These endeavors are enabling discoveries concerning the genetic basis of a myriad of complex traits and diseases. However, sifting through the constellation of SNVs to pinpoint pathogenic loci (regions and positions) is a challenging task. Genome-wide association studies (GWAS) have produced thousands of credible intervals of SNVs, which are frequently assumed to tag one or a few causal variants that influence the trait. In order to discover the underlying bona fide causal SNVs, researchers are increasingly adopting in silico functional analyses to prioritize candidate variants, either incorporating this information in the mapping algorithm a priori[2–5], or using it as a filter a posteriori in empirical studies[6–9]. Although it is clear that credible intervals in regulatory regions are enriched for functional elements[4,10–12], precise mapping of noncoding SNVs (ncSNVs) lags behind annotation of protein-coding variants.

Many computational tools have been developed to assess the functional impact of ncSNVs[13]. The general framework is to build predictive models that learn rules of combining multiple genomic annotations, functional attributes, and evolutionary features to discriminate pathogenic variants from non-pathogenic ones. During implementation, different assumptions of pathogenicity, various data and annotation resources, and assorted machine-learning and statistical algorithms have been employed. We summarize in Table 1 our survey of six current tools, namely CADD[14], CATO[15], DeepSEA[16], EIGEN[17], GWAVA[18], and LINSIGHT[19].

In particular, the definition of pathogenicity determines the class labels of ncSNVs in the training and/or testing steps (i.e., the positive class contains pathogenic variants and the negative class contains non-pathogenic variants). GWAVA defines pathogenic ncSNVs as disease-associated variants (DAVs) documented in the HGMD[20] or the ClinVar[21] databases, whereas non-pathogenic ncSNVs are those represented by common population polymorphisms (CPPs) in the 1000 Genomes Project[22]. Although this strategy works well for protein-coding variants, the small number of regulatory DAVs combined with potentially high false positive rates in the HGMD, and ClinVar annotations limits the generalizability of this strategy for noncoding variant diagnosis. CADD and LINSIGHT instead utilize deleteriousness inferred on evolutionary constraints as a proxy for pathogenicity. Since pathogenic variants are likely to depress fitness, measures of evolutionary selection are used to infer whether a site or an allele

has a deleterious effect on health (although it should not be assumed that the identified disease association is itself the cause of reduced fitness). Alternatively, molecular phenotypes, such as perturbation of chromatin structure or transcription factor binding, can serve as an indicator of potential pathogenicity and have been adopted by CATO and DeepSEA.

The attributes of an ncSNV that are potential predictors often include sequence-based features (e.g., motifs), evolution-based scores (e.g., conservation), summarized regulatory assay results (e.g., DNase hypersensitivity from ENCODE[23]), functional annotations (e.g. splice sites), and population allele frequencies. Since the function of a regulatory element is often tissue-specific, CATO further considers cellular context in its model. However, the high rate of missingness of cellular data has the consequence that CATO is unable to make predictions for a large number of ncSNVs. Given these training data, the relationship between pathogenicity and the various predictors of ncSNVs have been modeled using traditional statistical approaches (e.g., logistic regression) and advanced machine-learning techniques (both supervised an unsupervised learning).

Regardless of the distinguishing theoretical and empirical aspects of each method, these tools invariably claim excellent discrimination of pathogenic from non-pathogenic variants (reported AUROC values up to 0.97, Table 1), which has encouraged their application in over a thousand studies of the genetic basis of biomedical phenotypes. Interestingly, with the exception of GWAVA, the aforementioned methods did not report explicit cutoff scores to classify pathogenic and non-pathogenic variants. While ROC curves contrasting specificity and sensitivity are useful for evaluating the overall performance of a predictive model over the full range of the impact scores, the lack of recommended cutoff scores preclude assessment of traditional accuracy metrics. Users have to rely on ranking or arbitrarily determined cutoff scores to classify candidate variants. Consequently, the interpretation of the predictions is largely subjective. Furthermore, AUROC is insensitive to class imbalance (i.e. deviation of the ratio of positive and negative samples from 1:1)[24]. In most empirical studies, researchers aim to identify a small number of pathogenic variants among a relatively large number of non-pathogenic variants. Unfortunately, in these cases, high AUROC values do not imply high precision.

Notably, the concordance of pathogenicity predictions made by current tools is low[25] and in vitro evidence is frequently at odds with in silico assessments[14]. These observations suggest that the accuracies reported during the development of predictive methods

### Table 1 Properties of predictive models for six tools

| Method | Assumption of pathogenicity | Predictors | Modeling approaches | Performance (AUROC)[a] |
|---|---|---|---|---|
| CADD | Evolutionary fitness | Evolutionary parameters, ENCODE summaries, functional annotations, population frequencies | Support vector machines | 0.92[b] |
| CATO | Molecular functions | Cell type- and tissue-specific assays, evolutionary parameters, functional annotations | Logistic regression | NA[c] |
| DeepSEA | Molecular functions | Local sequences, evolutionary parameters | Deep learning, Logistic regression | 0.85 |
| EIGEN | None[d] | Evolutionary parameters, ENCODE summaries, population frequencies | Unsupervised learning | 0.79 |
| GWAVA | DAVs vs. CPPs | Evolutionary parameters, ENCODE summaries, population frequencies | Random forests | 0.97 |
| LINSIGHT | Evolutionary fitness | Evolutionary parameters, ENCODE summaries, functional annotations | Generalized linear model | 0.96 |

AUROC = area under the receiver operator characteristic curve, DAV = disease-associated variant, CPP = common population polymorphism
[a]Highest AUROC values in classifying DAVs and CPPs reported in the original publications
[b]CADD reported AUROC values that mixed coding and noncoding variants
[c]CATO predicts transcription factor occupancy instead of pathogenicity
[d]EIGEN uses an unsupervised learning approach and thus makes no assumption of pathogenicity during training

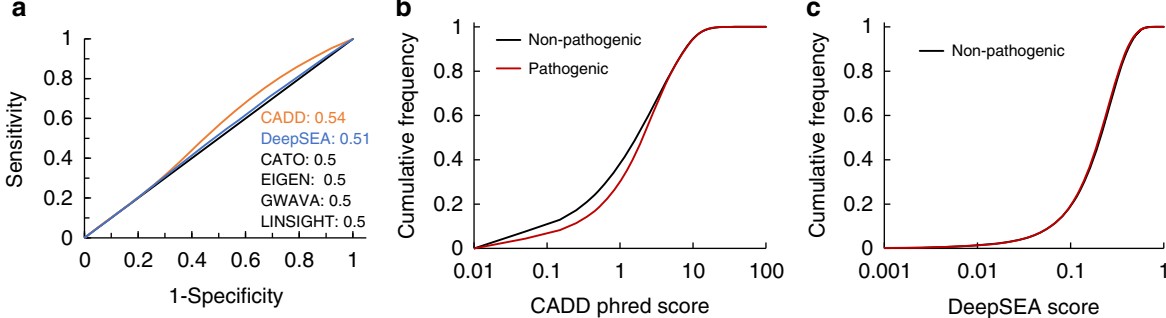

**Fig. 1** Performance of tested methods on detecting pathogenic variant in position-matched ncSNVs. **a** ROC curves with AUROC values displayed for each method. **b** Cumulative distribution of CADD scores. **c** Cumulative distribution of DeepSEA scores. Pathogenic ncSNVs were limited to variants not observed in human or any of the 57 non-human placental mammals

may not translate into high success rates when applied to fundamental empirical biological research questions. To test this proposition, we evaluate the performance of the aforementioned six predictive tools for three major biological tasks that represent common applications of pathogenicity predictions: distinguishing which of the four alleles at a causal site are pathogenic and which are protective (with implications for inferring how natural selection shapes variation at the locus[26]), fine mapping the causal variant within a credible interval of physically proximate ncSNVs that have similar statistical evidence for association[27–29], and ranking of candidate variants across the genome for prioritization of the likely disease-promoting gene(s). To avoid biases due to unknown cutoff values and class imbalance, we use relative ranks of pathogenic and non-pathogenic variants in balanced test sets as the performance metrics. We then explicitly evaluate the influence of class imbalance on all the predictions. Our results show that the existing methods have underwhelming performance with respect to the three discriminatory tasks, and point toward needed areas for improvement.

## Results

**Task 1: Discriminating pathogenicity of alternative alleles.** Given a genomic position with segregating alleles in a population, except in rare instances of balancing selection, one allele will usually be more deleterious than the other, presumably contributing to pathogenesis[30–33]. At any position, apart from the reference allele, we need to know which of the other three bases are pathogenic, because contrasts of their allele frequencies in different populations and/or relative to the ancestral state can help to elicit the direction of natural selection and shed light on the genetic basis of health disparities[34–36].

To evaluate the methods' performance for task #1, we reasoned that the strongest comparison likely to distinguish pathogenicity of ncSNVs at the same position by computational prediction would contrast variants that are commonly observed in human populations, but have no reported association with human diseases or quantitative traits (which we call non-pathogenic) with variants not observed in any of the species closely related to humans (which we call pathogenic). An evolutionary approach to assembling pathogenic variants was taken by the developers of CADD, who used simulated de novo mutations to represent such variants. CADD's mutations are SNVs that are different from the chimpanzee–human ancestral alleles and different from the human reference allele, hence presumably would be under purifying selection. In our study, we extended this concept by considering evolutionary history beyond human–chimpanzee and used 57 diverse non-human placental mammals (see the Methods section). Across the collection of these mammals,

there has been ~2.9 billion years of evolution, which ensures that a vast majority of sites in the genome have been evolutionarily tested on multiple mutations, given that the mutation rate is of the order of $10^{-8}$–$10^{-9}$ per base per year[37,38]. Following the strategy adopted by CADD and LINSIGHT using evolutionary principles to identify pathogenic alleles, and adopting the extensive taxonomic span considered here, we inferred that alleles never observed in any of these species are typically evolutionarily forbidden alleles that have been consistently subject to purifying selection due to their deleteriousness (they are pathogenic in a broad sense regardless of association with disease).

For the non-pathogenic variant set assembled in our tests, CPPs were used just as they were in the development of CADD, DeeepSEA, EIGEN, GWAVA, and LINSIGHT. At these positions, we further required that the derived alleles have frequencies between 5 and 15% in the human populations to minimize the influence of variants potentially under positive selection or balancing selection. We also tested whether the performance varied when we applied different population frequency cutoffs of derived alleles (see the Methods section).

With these criteria, we were able to assemble a position-matched balanced allelic test set of 55,453 positions. At each position, we identified a pair of pathogenic (i.e., evolutionarily forbidden) and non-pathogenic (i.e., CPP) alleles (see the Methods section). Our expectation was that a pathogenic allele would receive a significantly higher impact score (as defined for each of the six tested methods) than a non-pathogenic allele at the same position. Instead, we found that these methods were unsuccessful at this task. In fact, four of them (LINSIGHT, EIGEN, GWAVA, and CATO) reported identical scores for all alternative alleles at every position as they were not designed for allelic contrasts (Fig. 1a). Two methods (CADD and DeepSEA) produced different scores for pathogenic and non-pathogenic alleles at a position, but the AUROC was only slightly higher than 0.5 (AUROC = 0.54 and 0.51, DeLong test $p$-value = $10^{-11}$ and $10^{-13}$, respectively). Despite the statistical significance, the small effect sizes implied that the biological usefulness of these methods for task #1 is extremely limited, which was further confirmed by the statistically significant yet largely overlapping distributions of impact scores for pathogenic and non-pathogenic alleles (Wilcoxon signed-rank test $p$-value = 0, Fig. 1b, c). Similar results were also observed when pathogenic alleles were sampled only from positions that were completely conserved across all 58 mammalian species analyzed and contained no alignment gaps (AUROC = 0.49 and 0.57 for CADD and DeepSEA, respectively; see the Online Methods section and Supplementary Figure 1).

Therefore, it is apparent that none of the six methods are suitable for distinguishing among the alleles segregating at a site with respect to deleteriousness.

**Task 2: Resolving the identity of causal alleles.** Since GWAS typically resolves associations to credible intervals that may contain from a few to a hundred or more candidate SNVs[39,40], it has been proposed that algorithmic incorporation of functional and evolutionary scores might resolve true causal variants[27–29]. We thus next examined if the six prediction methods can distinguish pathogenic from non-pathogenic positions within credible intervals defined by physical distance or linkage disequilibrium. To simulate fine-mapping studies, we constructed a region-matched test set containing known pathogenic and non-pathogenic variants within certain genomic distances. As pathogenic variants, we used a collection of 764 DAVs that was a subset of 2037 HGMD variants, where we applied three filters that corresponded to one strong and two moderate criteria of pathogenicity in the clinical guidelines published by the American College of Medical Genetics and Genomics (ACMG)[41]. Even though ACMG guidelines are currently recommended only for defining pathogenic variants in the coding regions, we used them in an effort to enrich our collection with truly pathogenic variants (see the Methods section). To assemble a collection of putatively non-pathogenic variants, we used all noncoding CPPs with MAF > 5% that were located within the 1000 base pairs (1kbps) region surrounding the anchor DAV. Only those CPPs with no known disease associations were chosen. We then paired a DAV with each of its nearby CPPs to construct a region-matched balanced test set (see the Methods section). On average, there were 4.3 CPPs per anchor DAV, and this test set consisted of 3298 such pairs.

Between each pair of a DAV and a nearby CPP, a successful method should assign a higher impact score to the DAV. However, we were unable to reject the null hypothesis for CATO when the distance between a DAV and a CPP is less than 1 kbps, for EIGEN when distance <100 bps, for GWAVA when distance <50 bps, and for CADD, DeepSEA, and LINSIGHT when distance <10 bps (paired one-sided $t$-test $p$-value > 0.05). Even when the statistical tests reached the significance threshold, the effect size represented by the score difference between a pair of DAVs and CPPs was usually very small (Fig. 2a). For example, the DeepSEA score of a DAV, on average, was only 0.02 higher than its nearby CPP, while the full range of the DeepSEA score is between 0 and 1. Among all tested methods, CADD and LINSIGHT had relatively large positive components. However, even for these two methods, less than a quarter of the evaluations are greater than a standard deviation unit.

As the physical distance increases, the differences between the impact scores for DAVs and CPPs generally become more evident, i.e., pathogenic variants can be distinguished from non-pathogenic variants when they are located further apart (Fig. 2a). These results are likely explained by the high correlation between the impact scores of DAVs and matching adjacent CPPs, which decreases with increasing physical distance (Fig. 2b). For different tests, this may reflect contributions of evolutionary history of the haplotype block yielding similar conservation patterns, or the likelihood that closely linked alleles all lie within an extended regulatory stretch of chromatin with similar epigenetic marks.

A similar analysis was performed by defining vicinity with respect to linkage disequilibrium (LD) blocks with $r^2$ thresholds ranging between 0.8 and 0.99. Unlike physical distances, degree of genetic linkage does not affect the performance of CADD, CATO, EIGEN, and LINSIGHT significantly (ANOVA $p$-value range from 0.27 to 0.79, Supplementary Figure 2A). For the other two

methods (DeepSEA and GWAVA), the influence was significant when $r^2 > 0.9$ (ANOVA $p$-value < 0.001), but disappeared when $r^2$ dropped below 0.9 (ANOVA $p$-value = 0.35 and 0.14). This was likely due to the similar distributions of the sizes of LD blocks irrespective of $r^2$ thresholds (Supplementary Table 1), which also explains the relatively stable correlation of impact scores (Supplementary Fig. 2B).

We also stratified the data by using the evolutionary conservation of the position where a DAV is found: ultra-, well-, and least-conserved categories (absolute evolutionary rate = 0, <2, and ≥2 substitutions/site/billion years, respectively[42,43]). As reported previously[19,44], the maximum success rate of ranking a pathogenic ncSNV higher than an adjacent ncSNV was achieved when the variant affected an ultra-conserved position, which is up to twice as good as that at the least-conserved positions (Fig. 2c).

Among different types of regulatory elements, the prediction methods work best in identifying pathogenic ncSNVs disrupting a promoter (Fig. 2d). This may be because promoter regions are among the best studied regulatory elements in the genome, and most of the tools provide quite good discrimination for up to 97% of true DAVs in promoters. The improved performance may be attributed to the conservation of promoters[45,46] reflecting the strong evolutionary component of the impact scores. Alternatively, there may be an ascertainment bias toward experimental evaluation of the candidate variant on the basis of ENCODE-related criteria. We must note that previous comprehensive comparisons[14–19] have rarely ruled out the alternate hypothesis that one of the adjacent site encodes the causal variant.

To test whether the above results are sensitive to the presence of false positive DAVs in our collection of pathogenic ncSNVs, we used various filters to control population frequencies in selecting HGMD variants (see the Online Methods section). We observed qualitatively similar patterns regardless of data source (1000 Genomes Project or gnomAD) or frequency threshold (1%, 0.1%, 0.01%, or 0%) used for filtering variants (Supplementary Figs. 3 and 4). Supporting this result, we did not observe a relationship between variant population frequency and predicted impact score (Supplementary Fig. 5). We also experimented with using the ClinVar database as an alternative data source, which confirmed our findings in using HGMD variants (Supplementary Fig. 6). We also noted that DAVs are not evenly distributed across genes. Some genes, such as CCDC107 and HBB, have more than 40 DAVs within their 1 kbps upstream regions. To test if these clustered ncSNVs affect the performance, we randomly chose one DAV from each gene when available and repeated the analyses. Similar, but slightly worse, results were attained (Supplementary Fig. 7).

**Task 3: Prioritizing loci for gene set enrichment/prediction.** In order to identify gene sets enriched for contributions of multiple loci, or to assemble polygenic risk scores for prediction of disease risk, researchers often rank loci by genome-wide test statistics. Owing to the low power of GWAS to discover variants with small effects[47], spurious false positives greatly outnumber bona fide disease loci in any list of candidate intervals that includes regions below the conservative genome-wide Bonferroni threshold[48]. Similarly, among all ncSNVs generated via statistical association tests in a credible interval, parsimony suggests that only one or a small number of sites will be truly causal, which leads to the expectation of a low signal to noise ratio in candidate lists. To simulate these situations, we constructed test sets by mixing varying ratio of CPPs (non-pathogenic) and DAVs (pathogenic) ranging from 1:1 to 100:1. To avoid the confounding of physically proximate variants, we included only ncSNVs located beyond 1

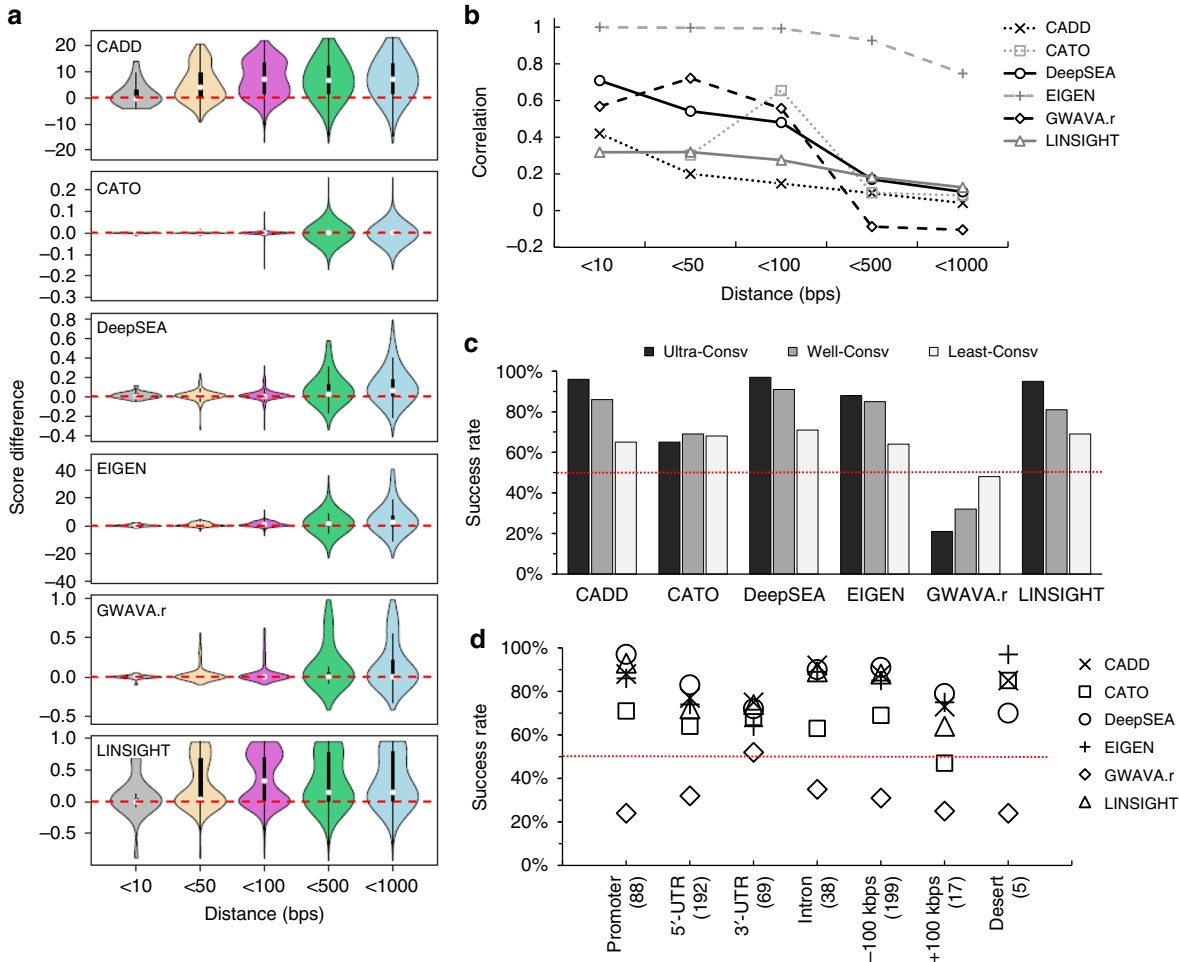

**Fig. 2** Performance of tested methods on region-matched pathogenic and non-pathogenic ncSNVs. **a** Violin plots show distributions of impact score difference between nearby pathogenic and non-pathogenic ncSNVs. Variants were grouped into bins based on distances measured in base pairs. Positive values imply that the pathogenic variant has the higher score. **b** Correlation of impact scores for ncSNVs located within given genomic distances. Pearson correlation coefficient values are displayed. **c** Success rates of ranking a pathogenic ncSNV higher than a non-pathogenic ncSNV located within 1000 bps vicinity. Data are stratified by conservation of the position harboring pathogenic ncSNVs. Since pathogenic and non-pathogenic variants are evaluated in pairs, the random expectation of success rates is 50% as represented by the red line. **d** Success rates stratified by the genomic context of pathogenic ncSNVs, including promoters, 5′- and 3′- untranslated regions (UTRs), introns, near-gene (+/−100 kbps) regions and gene-desert regions. GWAVA represents GWAVA region-matched scores

kbps from each other. For each test set, we ranked ncSNVs using impact scores and evaluated the ability to place pathogenic variants in the top 10 percentiles.

The expectation is that short-listed ncSNVs will be enriched with pathogenic variants, and indeed greater than 94% of ncSNVs in the top ranked predictions were from the pathogenic set when the test datasets consisted of non-pathogenic and pathogenic variants in equal numbers (mixing ratio = 1:1) (Fig. 3a). Since predictive models are usually trained using balanced datasets consisting of equal numbers of pathogenic and non-pathogenic variants[14–16,18,19], this result confirms that our test data are not unduly biased relative to those used to generate the impact scores.

In the more realistic scenario where the mixing ratio reaches 10:1, the short list contains approximately the same proportions of pathogenic and non-pathogenic variants, i.e., 50% false positives. As the mixing ratio of ncSNVs increases further, the discovery of the needle-in-the-haystack becomes progressively more difficult. The area under the precision-recall curve (AUPRC, Fig. 3b) provides a robust metric for unbalanced datasets, which has been a critical measure in assessing clinical diagnostic tests[24]. At the

desired AUPRC of 0.8, CADD and EIGEN are only suitable if >50% of the candidate ncSNVs in the collection are pathogenic, whereas DeepSEA and LINSIGHT retain some discrimination if >25% are pathogenic. GWAVA may be suitable up to a mixing ratio of 10:1. However, because 65% of our positive test data overlapped with the positive training data for the GWAVA model, its superior performance is likely an overestimation.

Performance is strongly a function of evolutionary conservation, since all tools performed better when the pathogenic ncSNVs were located at ultra-conserved positions. For example, LINSIGHT achieved AUPRC = 0.8 even when the ncSNV collection contained 64 times more non-pathogenic than pathogenic ncSNVs at ultra-conserved positions, although it performed poorly when the pathogenic ncSNVs were at least-conserved positions (Fig. 3c).

We also found that the performance varied with genomic context (Fig. 3d). Pathogenic ncSNVs in promoter regions are most likely to be correctly prioritized by DeepSEA, GWAVA, and LINSIGHT, while CADD works better on prioritizing pathogenic ncSNVs in introns. Interestingly, all tools are even more likely to

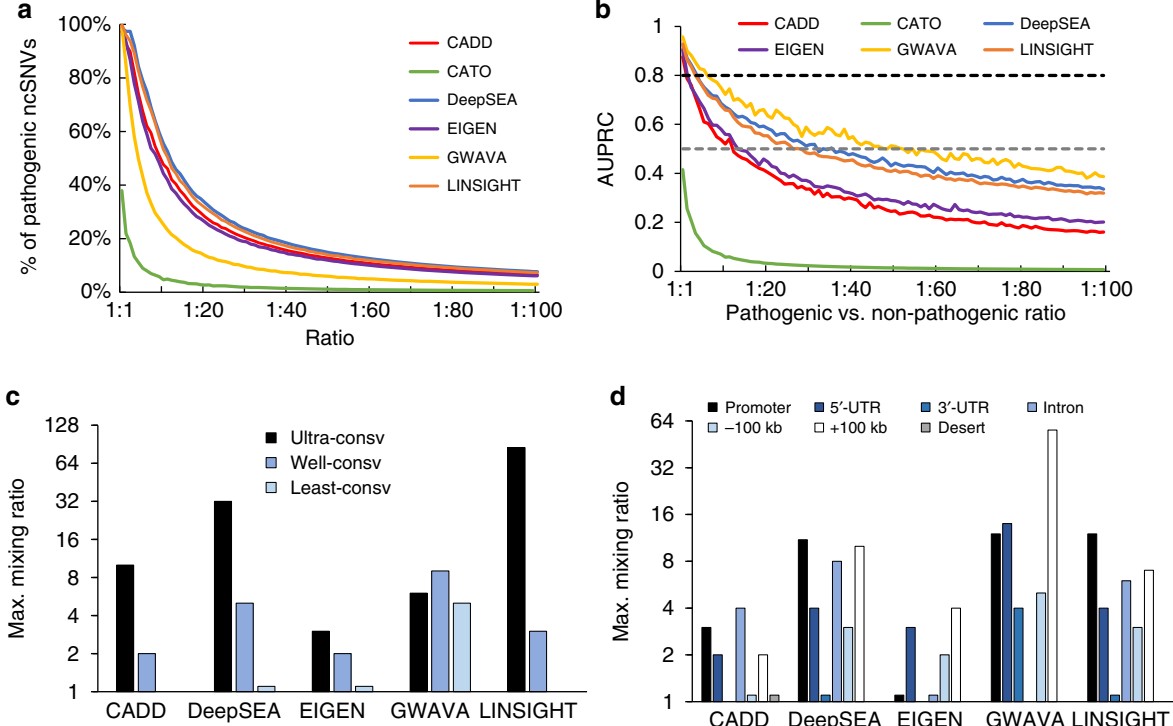

**Fig. 3** Performance of tested methods on prioritizing GWAS hits. **a** Fraction of pathogenic ncSNVs in the top 10 percentile of impact scores declines exponentially as the mixing ratio increases. **b** AUPRC value decreases as the mixing ratio increases. The gray dotted line presents AUPRC = 0.5 corresponding to random predictions. The black dotted line presents AUPRC = 0.8 corresponding to a desired performance in practice. **c** Maximum mixing ratio for each tool to achieve AUPRC > 0.8 when pathogenic ncSNVs disrupt ultra-, well-, or least-conserved positions. **d** Maximum mixing ratio for each tool to achieve AUPRC > 0.8 when pathogenic ncSNVs are inside different types of genomic regions. Since the highest AUPRC of CATO in any category was lower than 0.8, we did not include it in panels c and d. In this task, we used GWAVA unmatched scores

identify pathogenic ncSNVs in gene downstream regions than in upstream regions (+100 kbp versus −100 kbp). Two-way stratification of genomic context and evolutionary conservation showed that pathogenic ncSNVs in gene downstream regions were significantly enriched at ultra-conserved positions as compared to those in gene upstream regions (odds ratio = 3.1, Fisher's exact test p-value = $6 \times 10^{-5}$). We again tested if these patterns changed with varying stringencies of defining pathogenic ncSNVs and with gene clustering patterns, yielding similar findings (Supplementary Figs. 8-10).

## Discussion

With the realization that the majority of GWAS association signals lie in regulatory noncoding DNA, computational methods have quickly become indispensable for interpretation of the pathogenic propensities of ncSNVs with respect to their contributions to complex traits and human health[2–9]. Our assessment finds that six commonly used predictions are not suitable for discriminating between alternative alleles at the same position, and that they have a strong bias towards producing similar (pathogenic) impact scores for closely-located pathogenic and non-pathogenic ncSNVs. The lack of power to distinguish among alleles at the same position is likely because the tools were not designed to capture the characteristics of different alleles at the same position or closely-located positions. By contrast, we do observe some resolution of current methods at the regional level, particularly in the promoters and at ultra-conserved sites. These methods also have limited power to prioritize pathogenic ncSNVs when the proportion of pathogenic relative to non-pathogenic ncSNVs in a candidate list is small. Table 2 summarizes the

strengths and weakness of each method on the three biological tasks. In light of these patterns, low concordance of predictions made by current tools and the discordance between in vitro and in silico results is understandable[14,25].

A major component of the lack of predictive power for task #2 seems to be the high correlation between parameters characterizing sites located within at least a few hundred base pairs (Fig. 4). For example, CADD uses 63 parameters to build its model[14], all of which show high correlation with each other, many over a distance of 1 kb or more. Genomic features measuring sequence complexity and ENCODE features measuring functional features of chromatin[23] have the lowest discriminative power, either because the assays do not always resolve to a few nucleotides, or because the features aggregate signals covering tens of base pairs. As might be expected, motif scores are discriminative for ncSNVs separated by more than 10 bps. DNA structure features show the lowest correlation across all distance categories, but we found them neither to show consistent directional effects, nor to contribute much to pathogenicity predictions. Among evolutionary features, PhyloP[49] and PhastCon[50] scores computed using primate, mammalian, or vertebrate sequence alignments are also sensitive to the distance between sites, but GERP[51] scores are not.

Considering the potentially high ascertainment errors in labeling pathogenic vs. non-pathogenic ncSNVs, modeling techniques that are robust to noise in training data may offer potential gains in predictive accuracy. An alternative approach is to develop a composite score that may improve upon individual methods. We examined one such method, namely PRVCS[52], which unfortunately had poor performance (Supplementary Figure 11). Meanwhile, new high-throughput functional in vitro

**Table 2 Performance of six tools on three biological tasks**

| Method | Task 1[a] | Adj[d] | Task 2: Positional diagnosis[b] | | | | | | | Task 3: Diagnosis with noisy background[c] | | | | | | |
|---|---|---|---|---|---|---|---|---|---|---|---|---|---|---|---|---|
| | | | UC | WC | LC | Pro | 5'U | ups | int | UC | WC | LC | Pro | 5'U | ups | int |
| CADD | o | >10 bp | ++ | ++ | + | ++ | + | ++ | ++ | ++ | + | − | + | + | o | + |
| | | | + | | | | | | + | | | | | | | |
| CATO | − | >1 kbp | + | + | + | + | + | + | + | − | − | − | − | − | − | − |
| DeepSEA | o | >10 bp | ++ | ++ | + | ++ | ++ | ++ | ++ | ++ | + | o | ++ | + | + | + |
| | | | + | + | | + | | + | + | + | | | | | | |
| EIGEN | − | >100 bp | ++ | ++ | + | ++ | + | ++ | ++ | + | + | o | o | + | + | o |
| GWAVA | − | >50 bp | − | − | − | − | − | − | − | + | + | + | ++ | ++ | + | − |
| LINSIGHT | − | >10 bp | ++ | ++ | + | ++ | + | ++ | ++ | ++ | + | − | ++ | + | + | + |
| | | | + | | | + | | | | + | | | | | | |

Pathogenic variants are defined by location in ultra-conserved (UC), well-conserved (WC), or least-conserved (LC) intervals, or by location in the promoter (pro), 5'UTR (5'U), upstream gene region (ups) or intron (int)
[a]− indicates pathogenic scores are not specific to alleles at the same position, o indicates allele-specific scores but with low discriminative power
[b]success rates are indicated by: − (<50%), o (50–60%), + (60–80%), ++ (80–90%), +++ (>90%)
[c]non-pathogenic vs. pathogenic ratios are indicated by: − (<1:1), o (1–2:1), + (2–10:1), ++ (10–20:1), +++ (>20:1)
[d]Adj refers to adjacency corresponding to the minimum distance between pairs of pathogenic and non-pathogenic variants for which significantly different scores are produced in Task 2

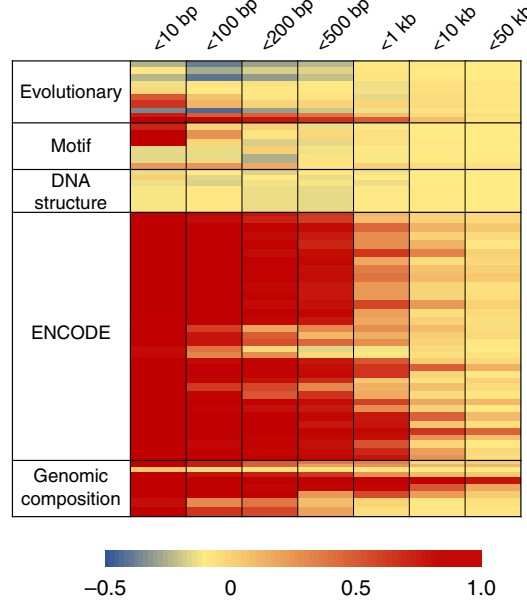

**Fig. 4** Correlation of ncSNVs located within specific genomic distances. Given a pathogenic ncSNV, all variants located within 10 bps, 100 bps, 200 bps, 500 bps, 1000 bps, 10,000 bps, and 50,000 bps of its flanking region were retrieved. For each group, the average correlation coefficient was computed for a predictor value and colorized according to the legend bar. A total of 63 predictors were organized into five categories. Evolutionary predictors include 10 scores computed by GERP, PhyloP, and PhastCon. Motif predictors include six similarity scores to known transcription factor binding sites. DNA structure predictors include four scores predicting nucleotide secondary structure. ENCODE predictors include 34 scores from UCSC regulatory tracks. Genomic composition predictors include nine scores of sequence complexity

assays have shown positional and even allelic resolutions. For example, massively parallel reporter gene[53] and CRISPR/Cas9[54] mapping are beginning to quantify the expression changes due to all possible regulatory sites in a credible interval. When these datasets accumulate to cover a significant fraction of the human genome and tissue types, we expect computational methods integrating this information in the model will have improved performance.

On the other hand, our results may be sensitive to the definitions of pathogenicity and also to the presence of false positive DAVs in the regulatory mutation databases. In this study, we chose computational and experimental comparisons of ncSNVs that are likely to be most discriminative given the current evidence. Varying inclusion/exclusion stringency and data source did not change the overall patterns (Supplementary Figures 3–10). At the same time, we find that the six methods perform well in task #2 for DAVs disrupting promoters or ultra-conserved positions, which may reflect a higher proportion of true positives as anchors. In this case, the reduced success rates in other regions could be interpreted as an artifact of ascertainment bias, although we consider this unlikely as a general explanation because DAVs in our test sets were accompanied with in vivo or in vitro evidence.

Alternatively, the performance of current tools may actually be worse than we describe, because we used DAVs implicated in heritable diseases as pathogenic variants for testing. These are generally of stronger functional impact than many ncSNVs that underlie complex diseases and traits. Indeed, adopting a measure of function expected to be only mildly correlated with pathogenicity, namely expression quantitative trait loci (eQTLs), confirms a high false negative rate for all six methods. Tested on 429 of the most significant functional variants causing the largest changes of gene expression in the GTEx dataset[55] (eSNVs), even the best-performing method failed to make correct predictions in 77% of the cases (Fig. 5a, Supplementary Table 2). The concordance among all the tools was again low. Even when we took a union of eSNVs predicted to be functional by at least one of the six methods, the collective sensitivity was less than 50% whether considering the peak effect size or p-value, individually, as evidence for causality. Therefore, current tools may not perform well for highly complex traits and diseases.

Overall, the strong tendency of each of the tools to assign more severe impact scores to ncSNVs found in evolutionarily conserved positions means that current approaches are highly biased toward selecting conserved sites at the expense of causal variants less constrained by evolutionary history (Fig. 5b). Among the functionally implicated eSNV alleles, more than 95% are found in non-conserved regions (phastCons score <0.5), implying as much as 20-fold underestimation of the number of causal ncSNVs that are not found in evolutionarily conserved regions.

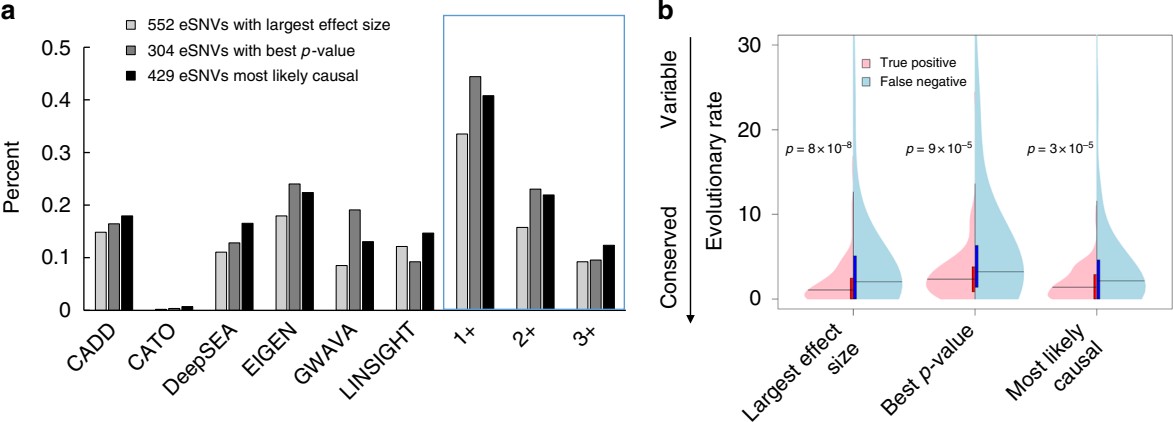

**Fig. 5** Performance of six methods for diagnosing eSNVs. In order to determine cutoff scores beyond which GTEx variants are classified as non-neutral, we selected the score cutoff that maximized balanced accuracy (TPR + TNR) on the DAV-vs-CPP dataset for each tool (Supplementary Table 2). **a** Proportion of eSNVs predicted as functionally non-neutral by each method and by at least one (1+), two (2+), or three (3+) methods. **b** Violin plots showing distributions of positional conservation, as measured by evolutionary rate. True positives are eSNVs predicted as functionally non-neutral by at least one method. False negatives are eSNVs predicted as functionally neutral by all six methods. Student *t*-tests were performed to compare evolutionary rates between true positive and false negative samples in each set. *p*-values are displayed

The most straightforward interpretation of our findings is that current approaches are not yet sophisticated enough to prioritize true causal variants. A contributing factor may be that functional effects are often context-dependent, as seen by the fact that eQTL are increasingly understood to vary across stimulation conditions[56–58]. For this reason, we included CATO[15], a method that incorporates cell-type specific epigenetic annotations in its predictive model, in our evaluations. But CATO does not have superior performance by our metrics. Newly discovered features, such as local regulation by lncRNAs or promoters also acting as enhancers[59], continue to suggest that novel features of gene regulation remain to be uncovered. On the other hand, Kircher carefully considered interaction effects in their machine-learning models[14], not finding any evidence for combinations of different types of marks to improve the CADD scoring. To the extent that closely located variants share conservation and epigenetic scores, fine resolution of eQTLs to one or two segregating sites may not be possible by computational criteria alone. Interference between the effects of multiple functional variants at a locus can even completely hide the nature of true causal variants for a non-trivial fraction of analyses[60].

A more controversial biological interpretation of the failure to discriminate pathogenic from non-pathogenic variants in close vicinity is that the parsimony model, which implicates only one or a few variants as causal for specific trait differences, is overly simplistic. On this interpretation, each eQTL (or GWAS) effect is actually the summation of very mild contributions from multiple variants in the interval. Such sites in high LD within a region of open chromatin bound by proteins that assemble an enhancer complex could all jointly contribute to the statistical signal. It is perhaps for this reason that simply stating the posterior probabilities for inclusion of a variant in the candidate list at an interval is more honest than promoting a single site, in the absence of appropriate experimental differentiation of the sites[39,61].

Our analysis clearly shows that the confounding of allelic, positional, and regional effects impedes the ability of current tools to correctly predict pathogenicity of ncSNVs, and provides valuable guidance to researchers who use these tools to prioritize regulatory variants for biological interpretation. We suggest that future improvements should be directed to deliberately

addressing the reasons for the high correlation of impact scores for closely linked SNPs, while also considering the impact of the proportion of false negatives in the candidate list on predictive accuracy.

## Methods

**Datasets**. Using the 1000 Genomes Project data, we randomly sampled 100,000 polymorphic noncoding positions that harbored minor alleles with a population frequency of between 5 and 15%. After filtering against the GRASP database[62] of GWAS associations in 2082 studies at association p-value of 0.05, 19,850 sites were excluded as potentially functional. At each remaining position, the allele with the highest observed minor allele frequency (MAF) was designated as non-pathogenic. Next, we checked if the remaining two alleles were reported as polymorphisms in the Great Ape Genome Project[63] and removed 644 positions. Then we searched the genome alignments[64] of 57 non-human placental mammals that collectively span ~2.9 billion years of evolution. Alleles absent in any of these mammals have been under persistent purifying selection due to deleteriousness and are presumably pathogenic. If two pathogenic alleles existed at the same position, we randomly chose one. If no pathogenic allele was found at a position, it was removed from further consideration. Using these criteria, we compiled the position-matched test set that consisted of 55,453 positions, each of which harbors a pair of pathogenic and non-pathogenic ncSNVs.

We also compiled a highly restricted subset of positions (75) in which pathogenic alleles were sampled only from positions that were completely conserved across all 58 species analyzed and did not contain any alignment gaps. This maximized our chances of sampling pathogenic alleles, because completely conserved positions are rare when species sampling is diverse and evolution is strictly neutral. Lindblad-Toh et al.[65] estimated this probability to be less than 2% for a set of 29 mammals. In our sequence alignment, the number of species is double the count in Lindblad-Toh et al.[65], so the completely conserved positions will occur with even lower probability. We estimated this probability by computer simulations using the 58 mammalian species phylogeny, a subset from the 100 vertebrate species phylogeny in the UCSC database[66], and species divergence times collected from the TimeTree resource[67]. We generated 100,000 neutral sites for many different neutral substitution rates ($10^{-8}$–$10^{-9}$ per base per year), G + C content biases (10–90%), and transition/transversion ratio[68] equal to 3.6 in the Pyvolve simulation library[69,70]. The fraction of sites with identical base across all 58 species, which would appear to be completely conserved in a sequence alignment, was determined from the datasets generated. For example, at most 0.6% of the positions were completely conserved when the mammalian substitution rate was $2.2 \times 10^{-9}$ per base per year[71] (Supplementary Fig. 12).

We also built two alternative test sets. In the first alternative set, we increased the upper limit of the population frequencies of the minor alleles to 95%, which resulted in a dataset containing 47,799 positions with matched pathogenic and non-pathogenic ncSNVs. In the second alternative set, we reduced the collection of non-human genomes to include 11 non-human primates, which resulted in a collection of 79,506 positions with matched pathogenic and non-pathogenic ncSNVs.

**DAV-vs-CPP comparison**. We used pathogenic variants annotated in the HGMD database (2015 version) as DAVs. To remove potential false positives, we applied three filters following the guidelines for the interpretation of sequence variants published by the American College of Medical Genetics and Genomics (ACMG)[41]. These guidelines catalogue four types of evidence of pathogenicity and recommend rules for combining these evidence types to diagnose the likely clinical significance of sequence variants. Our first filter removed HGMD variants labelled as DM that indicates some degree of uncertainty, and removed HGMD variants labelled as DP that indicates disease-associated variants but with no functional evidence (corresponding to the strong evidence PS3 in the guidelines). The second filter removed HGMD variants located outside known regulatory elements (corresponding to the moderate evidence PM1). The third filter removed HGMD variants observed in the 1000 Genome Project populations with >1% frequencies (corresponding to the moderate evidence PM2). We obtained 764 DAVs out of 2037 HGMD variants. As alternate filters, we further removed HGMD variants found in the gnomAD database (http://gnomad.broadinstitute.org/) with >1%, >0.1%, >0.01%, and >0% frequency, which resulted in 732, 648, 578, and 487 variants, respectively. The complete list of HGMD variants passing these filters are available in Supplementary Table 3.

Considering that the HGMD database collects DAVs from published studies showing segregation between cases and controls (corresponding to the strong evidence PS4), our filtering criteria meet or exceed the requirements to diagnose pathogenic variants in the ACMG guidelines. In total, we retained 764 HGMD DAVs as pathogenic ncSNVs. Further characterizing our selected set of DAVs, we found that 32 had a population frequency greater than 1% in the gnomAD database, 513 were within 1 kb of a transcription start site (due to our selection criteria of DAVs in known regulatory elements that are enriched in near-TSS regions), 20 out of 46 intronic variants were within 1 kb of an exon junction, and 238/174/350 were at ultra-/well-/least-conserved sites, respectively. For each DAV, we searched within its 50,000-bps flanking region for common population polymorphisms (CPPs) that had a global MAF greater than 5% in 1000 Genomes Project. We retrieved 87,811 CPPs and designated them non-pathogenic. Linkage disequilibrium (LD) blocks were defined by $r^2$ value as released by the 1000 Genomes Project.

We also built two alternative test sets by varying criteria of pathogenic ncSNVs. In the first alternative test set, we used the ClinVar database[21] instead. After removing coding variants and variants with population frequency >1%, we obtained 272 ClinVar DAVs as pathogenic. In the second alternative test set, we chose one pathogenic variant per gene. Specifically, we identified the closest gene of each DAV based on the RefSeq annotations. Seven hundred fifty (98%) of the 764 DAVs are within10 kbps of 306 RefSeq genes. We randomly chose one DAV in the10-kbps flanking region of a gene and constructed a gene-balanced test set.

**GTEx variants**. We retrieved single-tissue eQTL datasets from https://www.gtexportal.org/home/datasets that contains expression change information regarding significant SNP-gene associations for 1,922,134 mutations at 1,921,848 unique positions across 44 different tissues. For each of these positions, we retrieved the phastCons primate conservation scores from the UCSC database (ftp://hgdownload.soe.ucsc.edu/goldenPath/hg19/phastCons46way/primates/). We identified the best functional candidates for analysis by first selecting 1000 SNPs with the largest effect size (change in gene expression, $|\beta|$) that were deemed to have the best association (annotated as chosen by GTEx, based on permutation testing probability) with the gene they regulated. This set contained many eQTLs that strongly affected multiple genes across several tissues, and were thus present as duplicates. Removing 571 such duplicates provided 429 unique positions that serve as the strongest candidates of functional activity in the GTEx catalog, i.e., they confer the largest expression change for one or more genes, across one or more tissues, with the most confidence. Less stringent criteria for putatively causal variants provided additional datasets: SNPs with 1000 lowest association $p$-values and 1000 largest effect sizes. Removing duplicates from these provided 304 and 552 SNPs in the best $p$-value and largest effect size datasets, respectively. Some scores were not available for all variants (see Supplementary Table 4).

**Prediction tools**. For CADD, we downloaded the full set of impact scores and annotations, for every possible SNV, from http://cadd.gs.washington.edu/download, and used the tabix program[72] to extract the relevant data. For variant effect prediction we used the CADD phred scores. CADD scores were available for all the SNVs. For EIGEN, we downloaded the set of scores for all genomic positions from https://xioniti01.u.hpc.mssm.edu/v1.1/, and used the tabix program to extract the relevant positions. For variant effect prediction we used the EIGEN-PC Phred scores, because they produced the highest AUROC value on the DAV-vs-CPP dataset among the other three EIGEN scores. EIGEN scores were available for 690 DAVs and 82,825 CPPs. For LINSIGHT, we downloaded the full set of pathogenicity scores for every possible SNV, from http://compgen.cshl.edu/~yihuang/tracks/LINSIGHT.bw. LINSIGHT scores were available for 739 DAVs and 87,692 CPPs. For DeepSEA, we queried scores from the online interface at http://DeepSEA.princeton.edu/job/analysis/create/. DeepSEA scores were available for all SNVs. For GWAVA, we downloaded the set of impact scores from ftp://ftp.sanger.ac.uk/pub/resources/software/gwava/v1.0/annotated/gwava_db_csv.tgz for all known population polymorphisms. For variant effect prediction, we used the score

computed by the region-matched model for tasks 1 and 2, and the score computed by the unmatched model for task 3 and for eQTL prioritization. For our DAV-vs-CPP dataset, scores were available for 246 DAVs, and 86,744 of the 87,811 CPPs. For CATO, we downloaded the scores for all SNPs overlapping DNase I–hypersensitive sites from http://www.mauranolab.org/CATO/dbSNP142.CATO.V1.1.txt.gz. For our DAV-vs-CPP dataset, scores were available for 155 DAVs, and 17,804 CPPs.

**Calculation of evolutionary rates**. We estimated the rate of evolution for each position by using the Fitch algorithm applied to nucleotide sequence alignments of 57 placental mammalian species from the UCSC Genome Browser[73,74]. This produced an absolute substitution rate for each nucleotide position in the unit of substitution per site per billion years. We also used regional phastCons scores available from the UCSC browser.

**Code availability**. All scripts used to extract variants from databases, and to perform computations, are available from the authors upon request.

## Data availability
All datasets and identifiers are available for download from mypeg.info/datasets.

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

## Acknowledgements

We are grateful to Urko Marigorta, Biao Zeng, Ruoyu Tian and colleagues from the Gibson and Kumar labs for their support and helpful comments. This research was supported by NIH grant 1-R01-HG008146-02.

## Author contributions

S.K., L.L., and G.G. conceived and designed the study. L.L., M.S., and R.P. carried out the statistical analyses, with assistance from R.P. and P.C. in accessing and generating datasets and scores, and participated in the conception and design of the study. L.L., S.K., and G.G. drafted and revised the manuscript and all authors interpreted the data.

## Additional information

**Competing interests:** The authors declare no competing interests.

