## [Peer Review File · Nature Communications]

Reviewer #1 (Remarks to the Author):

Liu et al. present an assessment for the predictive power of various published methods/scores with regards to pathogenicity of non-coding regions. Six popular scores are discussed in the manuscript: CADD, DeepSEA, LINSIGHT, EIGEN, GWAVA and CATO and their performance is analysed at three levels: allelic, positional and regional. The authors raise noteworthy concerns about the limitations of existing methods towards sufficiently and accurately pin-pointing pathogenic non-coding variants. Their results show that success rates are primarily proportional to conservation (e.g. higher in promoters). Additionally, all methods struggle to differentiate pathogenic non-coding variants from benign neighbouring variants. Moreover, these scores do not perform efficiently when dealing with a high imbalance of positive and negative data points (i.e. pathogenic vs. non-pathogenic elements), which is generally the case when working with genomic data.

In summary, the authors suggest instead that new methods should be developed relying less on conservation to help identify pathogenic variants that may have emerged irrespective of evolutionary constraints. They also suggest that utility of current scores should be more about quantifying a genomic interval instead of a single site.

Even though this work does not introduce a new method, the points underlined in this comparative review are important to be taken into consideration when using the examined methods in a study to claim/support pathogenicity of non-coding regions. Additionally, the authors justify and provide novel directions and insights towards developing better methods in the future and thus can be used as a reference in this endeavour. Overall, the manuscript is well written with clear figures for the presentation of their results. A few points are provided below.

Major

* Line 85: It would be interesting to see if a combination of the examined scores would better distinguish between pathogenic and non-pathogenic non-coding regions. Although we suspect there to be high correlation between features this will test the hypothesis that each score may not be sufficient on its own to make any distinction between pathogenic and non-pathogenic ncSNVs. However, a combined model might provide more discriminating power than individual scores, suggesting that each score captures part of the underlying information with regards to a region's pathogenicity propensity.

* Line 85: A statistical test for comparison between the examined ROC curves would be more concrete – e.g. with the DeLong et al. (1988) method (implemented in statistical software packages)

* Line 86: Mann-Whitney U (or other) test required to assess statistical significance of the difference reported between two distributions (CADD and DeepSea scores in pathogenic vs non-pathogenic).

* Line 107: The claim “too small to provide biological meaning” is vague, especially after mentioning that the statistical tests reached $P < 0.05$. Maybe alternative phrasing will be useful since observing only small ‘visible’ differences is pretty common when working with whole genome data.

* Line 116: Respective correlation scores (Pearson's r) needed.

* Line 120: “significantly” – P-values from the statistical comparisons between distributions for different R^2 values required.

* Line 127-8: “promoters are the most conserved across species”: citation needed. Additionally, it would be useful to include in this analysis other highly conserved regulatory elements (e.g. UCNEbase).

* Line 147: “predictive models are usually trained using balanced datasets”: is that actually the case for all the examined scores? Citation for those that this statement holds true would be helpful.

* Overall it would also be of importance to include commentary about using emerging OMICs technologies to score pathogenicity based on functional in vitro screens, e.g., the recent crisprQTL: <https://www.biorxiv.org/content/early/2018/05/04/314344>

Minor comments.

* Line 45: It would be helpful to have access to a table with representative accuracy scores attained by each of the examined scores/methods based on the original papers or other studies based on them, as a reference for comparison.

- * Line 90: identify -> identity
- * Figure 2: "gene-desert regions" are mentioned in the legend but are not shown in Fig 2D.
- * Figure 3b: Missing x-axis labels
- * Supplementary Table 2: Define "BACC" acronym

Reviewer #2 (Remarks to the Author):

This manuscript titled "Biological relevance of computationally predicted pathogenicity of noncoding variants" is an extremely important and insightful contribution to the literature. In this paper, Liu et al. compare and contrast a number of variant pathogenicity prediction tools in an effort to evaluate whether the accuracy rate translate into high rates of success in using these predictions for biological research. Overall, their findings are a not surprising, but sobering, realization that the current state of pathogenicity prediction algorithms are imperfect. There is considerable more work to be done in this space and they offer some insights into where these efforts are needed.

Some areas where I felt the manuscript could benefit from additional discussion or revision include the following:

- On page 4, the authors discuss the accuracies of these prediction methods. I think for the readers who are less familiar with these tools, a brief discussion of what this accuracy means - or how the authors of the tools intend for accuracy to be interpreted - would be useful. This accuracy is especially surprising given the statement on page 4, line 47-48, citing how in silico assessments are at odds with in vitro evidence. Again, this is not surprising to those who are experts in this field, but for the general readership, a bit more explanation would be helpful.
- On page 6, lines 85-88, it is a bit surprising that there is a complete overlap of pathogenicity scores for pathogenic and non-pathogenic ncSNVs; isn't this what these methods are intended to do? Differentiate between pathogenic and non-pathogenic?
- On page 7, the authors mention the use of the HGMD database for selection of the variants. I also looked in the methods on this topic. Was anything done to prohibit the number of false positive variants from HGMD - as HGMD is known to have a large number of false positive variants in the database. Was anything done in the selection of variants to minimize the number selected?
- On page 15, lines 244-246, the authors discuss that each eQTL effect could be the summation of minor contributions from multiple variants within an interval. I think that this is an interesting and likely reasonable explanation. I wonder if there is a way to test this theory in your simulations? Or perhaps expanding on this idea bit would be useful. I think as more people consider new pathogenicity prediction algorithms.

All in all, I think that this is a really important contribution to the literature. Too many researchers are using one of these types of tools and moving research forward based on these predictions, which we are learning are not very accurate. This series of analyses are providing a nice framework from which to move the field forward to develop better algorithms.

Reviewer #3 (Remarks to the Author):

In "Biological relevance of computationally predicted pathogenicity of noncoding variants", Liu et al assess six commonly used variant annotation tools in terms of their ability to differentiate sets of non-coding variants with or without disease associations. They conclude that these tools have minimal power to differentiate pathogenic from non-pathogenic variants within regulatory regions. I find that there are a few valuable points and interesting ideas in this manuscript, the general topic is an important one, and there is little doubt that new and improved annotations are needed to better discriminate phenotypically relevant non-coding variants. However, the core conclusions rest upon unrealistic, or at best unproven, assumptions. Details are below, but the high-level summary is that the criteria used to define the "pathogenic" and "non-pathogenic" test sets are

error-prone, and insufficient effort is made to account for these test set errors at either the bulk or individual variant level. These test set errors are likely to be large, but are at the very least not negligible, and would tend to make the annotations appear to perform more poorly than they actually do. As a result of these issues, and some other more minor ones, I do not find the conclusions drawn in the manuscript to be well supported. In fact, some of the results suggest the opposite conclusion. My more detailed comments will be structured within the three “biological tasks” that are used in the manuscript.

The first task studied is the ability for annotations to differentiate distinct alleles at a given genomic position. To do so, a set of “non-pathogenic” alleles is defined as those with a minor allele frequency between 5% and 15% that are absent from GWAS catalogs. A set of “pathogenic” alleles is then defined as those alleles at the same genomic position that are not observed in non-human primates, as these are “presumably deleterious (non-neutral)”. This presumption is essential to the analysis but not justified or critically examined. It is also the opposite of the null model, which is an assumption of neutrality. This is particularly true for SNV alleles at a genomic position that is known to carry at least one high-frequency allele and which therefore have a higher than background probability of being neutral. In light of this null, substantive evidence is required to infer that any given allele or set of alleles is not neutral. Towards that end, the analysis presented relies upon the absence of an allele from an unspecified number of primate assemblies that span an unquantified amount of evolutionary divergence. However, no estimation is provided as to how well such evidence actually differentiates deleterious from neutral alleles, especially among alleles where the prior probability of neutrality is particularly high. As an extreme example, consider a scenario wherein only the chimpanzee reference assembly is used; the absence of an alternative allele in this assembly has very little information content and nearly every possible allele would be labeled as “presumably deleterious”. Obviously, inclusion of more primates improves the signal, but even the complete repertoire of available primate assemblies is unlikely to have a high positive predictive value for inferring pathogenicity in this case. My belief is that the observed lack of separation by any of the annotations between the two sets of variants in this task is overwhelmingly due to a genuine lack of biological separation. Even if I am wrong, the burden of proof is on the authors to thoroughly evaluate their core presumption and the extent to which errors arising from that presumption affect interpretation of the results.

Other related questions and concerns:

- Why 5-15% MAFs to select “neutral” variants? What about other parameter choices? Why impose a maximum MAF?
- To what extent are the ncSNVs that are presumed neutral as a result of not being detected by a GWAS likely to include genuinely disease-associated variants that have simply not yet been detected? This is the converse to my main concern, and admittedly likely to have a smaller impact, but is another potential artifact that would narrow any genuine biological gap between the two variant sets.
- Many details on the selection of “presumably deleterious” alleles are missing. Which primates were analyzed? Were all ncSNVs tested, or only those at positions that were aligned to other primates? If not all primates were required to be aligned at that position, how many were required? Were primate population genetic datasets examined to look for within-species polymorphisms, or just the reference assemblies? Etc.

The second biological task is more credible than the first, but only mildly so. The basic concern in task one – i.e., that the variant set labeled as “pathogenic” is contaminated with a large but unestimated proportion of actually non-pathogenic variants – also holds for task two. In this case, pathogenicity is defined by presence in HGMD coupled to the existence of “demonstrated effects ... on in vitro transcription”. The latter assertion is not explained or qualified; i.e., what standards of regulatory effect size, statistical support, assay type, etc., were used to define “demonstrated effects”? More important is the issue of pathogenicity, which is more difficult to assess and for which evidence of transcriptional effect is neither necessary nor sufficient. The conclusion that these DAVs are actually pathogenic depends on the presumption that presence in HGMD equates to being pathogenic. However, this is not true, as the false positive rate of HGMD is substantial.

For example, Cassa et. al. (Human Mutation article in 2013) found that 8.5% of variants in HGMD are present in at least one individual in 1000 Genomes and Bell et. al. (Science Translational Medicine article in 2011) found that 27% of reported recessive variants were too frequent in a sample of only 104 people to be pathogenic. In a study of 1,000 individuals, Dorschner et. al. (AJHG article in 2013) observed 239 HGMD-flagged variants, finding that 30% of these are too frequent to be pathogenic and, conversely, that only 7.5% meet rigorous pathogenicity criteria. As yet another example, Paludan-Muller et. al. (Clinical Genetics article in 2017) showed that reported "pathogenic" variants for one disease sum to an ExAC prevalence of 1 in 132, but the disease associated with these variants has an actual prevalence of 1 in 10,000, indicating that the overwhelming majority of observed "pathogenic" alleles are either not pathogenic or have very low penetrance; this manuscript also found that several variants that had been explicitly published as non-pathogenic were flagged as "pathogenic" in HGMD. Thus, while the precise rate of errors across the database is unclear (owing to various ascertainment biases and incomplete knowledge in studies of true and false positive rates), many studies have established that HGMD has a large number of false positives, even among protein-altering alleles; reportedly pathogenic non-coding alleles, for which available allele-frequency catalogs are shallower and which have intrinsically lower prior probabilities of pathogenicity, are likely to have even higher false-positive rates. As such, HGMD presence is an insufficient source of evidence for pathogenicity. In fact, this has been codified into the American College of Medical Genetics clinical variant interpretation standards, which categorizes presence in a disease database as "supporting", the weakest of four levels of evidence; such information can only be used as one of multiple additional factors, including at least one much stronger piece of evidence, to infer pathogenicity. I find it likely that a large fraction of the "pathogenic" DAVs in task two are actually non-pathogenic and, as for task one, the results presented thus fail to separate predictor error from a genuine lack of biological differentiation between "pathogenic" and "non-pathogenic". And, again, as for task one even if I am wrong it is incumbent on the authors to justify their definitions and provide evidence that they accurately separate truly pathogenic and truly non-pathogenic alleles.

I note that the above concern is acknowledged in the manuscript but ignored without explanation. Specifically, the text beginning at line 127 explains that variants disrupting conserved promoters are more accurately distinguished by prediction methods and that such variants "may also be the most definitively defined in the HGMD database." Assuming that "most definitively defined in the HGMD database" means most likely to truly be pathogenic, I agree. The fact that the predictors more effectively separate conserved promoter DAVs from CPPs is, in my opinion, a direct result of the fact that conserved promoter DAVs in reality have a lower false positive rate with respect to disease relevance. In this scenario the predictors are producing the correct outcome, a result that contradicts the main conclusions of the manuscript.

Related questions and issues:

- There is a lot of missing information about the set of HGMD variants. What processing was applied to HGMD? What does the resulting collection of variants look like in terms of GnomAD/ExAC allele frequencies, distance to TSSs, distances to exon junctions, conservation, etc.? How many genes/loci are represented and how sensitive are the results to the fact that large fractions of HGMD alleles cluster near a small number of genes?
- Why use HGMD data from 2015 rather than 2018?
- Why not use ClinVar, a free, open-access, community-driven tool that provides more and better information pertaining to the pathogenicity of any given variant, including how many and which labs scored the variant, the standards used by the reporting labs, the scores assigned by each lab, etc.?
- The LD-based analyses on page 8 are not clear as written. I believe the r -sq thresholds are used to select the blocks of CPPs near to a DAV, regardless of the correlation between the DAVs and the CPPs per se. As written the suggestion is that the CPPs are selected as a result of pairwise correlations with the DAVs, which I do not believe is the case.

Task three depends on variant definitions from task two, and therefore suffers from the same problems.

Response to Reviewers' Comments and Suggestions

We sincerely thank all reviewers for their constructive remarks. In this revised manuscript, we have made two major types of revisions to address all the concerns raised by the reviewers. First, we have expanded the descriptions and analyses to provide more background information, justify the definitions of pathogenicity/deleteriousness, and explain the rationale for the tests conducted. Second, we have re-evaluated the disease-associated variants (DAVs) in the HGMD database after applying more stringent filters recommended by the American College of Medical Genetics and Genomics (ACMG) standards and guidelines for the interpretation of sequence variants, which slightly reduces the size of the test set but does not qualitatively alter the results. These results, along with the additional analysis of ClinVar data, as recommended by R3, clearly establish the generality and robustness of our conclusions.

The manuscript is substantially longer than the initial page-limited submission, reflecting the changes, and we have highlighted all the modified text in blue in the main file which includes the Figures as well as a new Table. Below we provide point-to-point responses to each of the reviewer criticisms, but before doing so wanted to make four general points, particularly in light of Reviewer 3's thoughtful comments on the difficulty of defining suitable test sets and pathogenicity scores.

- First, it was not our intention to claim that the methods are “useless”, and so we have been sure to highlight that they do tend to perform well in promoter regions and at ultra-conserved sites, while also explicitly stating that four of them failed in task #1 since they report the same score for all alleles at the same position.
- Second, the criteria of pathogenicity we use are based on current practices but with a higher stringency (see response *R2.d* and *R3.b-i*). It is designed to contrast neutral and deleterious variants as best as we can using state-of-the-art criteria. Hence the performance we report is likely more optimistic than the real performance, which means that methods will perform worse than we report. We take care to point out the limitations and assumptions of current trends.
- Third, R3 suggested that we further increase the stringency because the databases cataloguing pathogenic variants contain a relatively high level of false positives. We agree, and now use very stringent criteria, following clinical guidelines published by the ACMG (*R2.d* and *R3.d*).
- Fourth, the dependency of precision (positive predictive value) on the mixing ratio of true to false positives is now explicitly discussed in the text. While precision is a key criterion in evaluating clinical diagnostic tests, we feel that this point is under-appreciated in the genetics community, which usually relies on specificity and sensitivity alone to evaluate score performance.

It is possible that our results are affected by an imperfect collection of neutral/deleterious variants even after our careful curations (now discussed in the text). But if the biases are so strong that the overall patterns are distorted significantly, that will be an extreme assessment at odds with accepted current practices, and readers can make the judgement.

In summary, given high enthusiasm from two reviewers and many positive comments from Reviewer 3, it seems to us that the more standard interpretation is an important message that is worthy of publication in *Nature Communications* as it will have a high impact.

Sincerely,

Sudhir Kumar and Greg Gibson

Reviewer #1

Comment R1.a. Liu et al. present an assessment for the predictive power of various published methods/scores with regards to pathogenicity of non-coding regions. Six popular scores are discussed in the manuscript: CADD, DeepSEA, LINSIGHT, EIGEN, GWAVA and CATO and their performance is analysed at three levels: allelic, positional and regional. The authors raise noteworthy concerns about the limitations of existing methods towards sufficiently and accurately pin-pointing pathogenic non-coding variants. Their results show that success rates are primarily proportional to conservation (e.g. higher in promoters). Additionally, all methods struggle to differentiate pathogenic non-coding variants from benign neighboring variants. Moreover, these scores do not perform efficiently when dealing with a high imbalance of positive and negative data points (i.e. pathogenic vs. non-pathogenic elements), which is generally the case when working with genomic data.

Response: Thank you. This comment summarizes our results clearly and highlights their importance.

Comment R1.b. In summary, the authors suggest instead that new methods should be developed relying less on conservation to help identify pathogenic variants that may have emerged irrespective of evolutionary constraints. They also suggest that utility of current scores should be more about quantifying a genomic interval instead of a single site.

Response: Indeed, this is one important implication of our findings.

Comment R1.c. Even though this work does not introduce a new method, the points underlined in this comparative review are important to be taken into consideration when using the examined methods in a study to claim/support pathogenicity of non-coding regions. Additionally, the authors justify and provide novel directions and insights towards developing better methods in the future and thus can be used as a reference in this endeavor. Overall, the manuscript is well written with clear figures for the presentation of their results. A few points are provided below.

Response: Thank you.

Comment R1.d. Line 85: It would be interesting to see if a combination of the examined scores would better distinguish between pathogenic and non-pathogenic non-coding regions. Although we suspect there to be high correlation between features this will test the hypothesis that each score may not be sufficient on its own to make any distinction between pathogenic and non-pathogenic ncSNVs. However, a combined model might provide more discriminating power than individual scores, suggesting that each score captures part of the underlying information with regards to a region's pathogenicity propensity.

Response: This is an excellent idea. Some others have attempted to build a combined model. We identified a published method, namely PRVCS, which computes a composite score based on predictions of other methods. However, the performance of PRVCS was not better than individual tools. In fact, it performed worse at the allelic and positional levels and showed stronger biases towards conservation, which was counterintuitive. We have included these results in the Supplementary Figures 9 and 10. We feel that it would take considerably more time and work, beyond the scope of this study, to derive a better composite predictor.

Comment R1.e. Line 85: A statistical test for comparison between the examined ROC curves would be more concrete – e.g. with the DeLong et al. (1988) method (implemented in statistical software packages); Line 86: Mann-Whitney U (or other) test required to assess statistical significance of the difference reported between two distributions (CADD and DeepSEA scores in pathogenic vs non-pathogenic); Line 116: Respective correlation scores (Pearson's r) needed.

Response: These are good suggestions. We have now presented the results of corresponding tests to assess the statistical significance. Specifically, we used (i) the DeLong method to compare the ROC curves in task #1, (ii) the Wilcoxon signed-rank test to compare the scores of pathogenic and non-pathogenic alleles in task #1; we chose Wilcoxon signed-rank test instead of Mann-Whitney U test because the scores are

matched by positions, and (iii) the Pearson's r to quantify the correlations of scores of variants in linkage disequilibrium (see Supplementary Figure 2B).

Comment R1.f. Line 107: The claim “too small to provide biological meaning” is vague, especially after mentioning that the statistical tests reached $P < 0.05$. Maybe alternative phrasing will be useful since observing only small ‘visible’ differences is pretty common when working with whole genome data.

Response: We have updated the text.

Comment R1.g. Line 120: “significantly” – P-values from the statistical comparisons between distributions for different R^2 values required.

Response: We have now included the ANOVA results of comparing the multiple distributions of score differences of variants in linkage disequilibrium at various levels.

Comment R1.h. Line 127-8: “promoters are the most conserved across species”: citation needed. Additionally, it would be useful to include in this analysis other highly conserved regulatory elements (e.g. UCNEbase).

Response: We have added several citations that have suggested higher conservation levels for promoters than other regulatory elements (e.g., PMID: 15735639, 25635462). We explored the use of variants available from UCNEbase, but found that only one variant from our filtered HGMD set overlapped an ultra-conserved non-coding element in UCNEbase, which did not permit the inclusion of UCNEbase data in further analysis.

Comment R1.i. Line 147: “predictive models are usually trained using balanced datasets”: is that actually the case for all the examined scores? Citation for those that this statement holds true would be helpful.

Response: In machine learning, a best practice in building predictive models is to ensure class balance. For this reason, computational methods use over-sampling, down-sampling or sample weights to balance positive and negative sample sizes. Specifically, DeepSEA uses sample weights, and CADD, CATO, GWAVA and LINSIGHT use down-sampling. EIGEN employs an unsupervised approach, so their training data lack class labels. We have now mentioned this point in the revised manuscript.

Comment R1.j. Overall it would also be of importance to include commentary about using emerging OMICs technologies to score pathogenicity based on functional in vitro screens, e.g., the recent crisprQTL: <https://www.biorxiv.org/content/early/2018/05/04/314344>

Response: We agree with the reviewer that functional in vitro data are promising in assisting variant interpretation. We have added a paragraph discussing this emerging trend and referencing recent CRISPR and MPRA published papers (we prefer to avoid referencing yet-to-be-peer reviewed work).

Comment R1.k. Line 45: It would be helpful to have access to a table with representative accuracy scores attained by each of the examined scores/methods based on the original papers or other studies based on them, as a reference for comparison.

Response: We have now added Table 1 to show the AUC values of ROC curves of the six methods reported in their original publications, while also listing the different methods used to train and test their statistics.

Comment R1.l. A few edits: Line 90: identify -> identity; Figure 3b: Missing x-axis labels; Supplementary Table 2: Define “BACC” acronym

Response: We have made the corrections.

Comment R1.m. Figure 2: “gene-desert regions” are mentioned in the legend but are not shown in Fig 2D.

Response: We have now added a “Gene Desert” category to the figure.

Reviewer #2

Comment R2.a. This manuscript titled "Biological relevance of computationally predicted pathogenicity of noncoding variants" is an extremely important and insightful contribution to the literature. In this paper, Liu et al. compare and contrast a number of variant pathogenicity prediction tools in an effort to evaluate whether the accuracy rate translate into high rates of success in using these predictions for biological research. Overall, their findings are a not surprising, but sobering, realization that the current state of pathogenicity prediction algorithms are imperfect. There is considerable more work to be done in this space and they offer some insights into where these efforts are needed.

Response: Thank you.

Comment R2.b. - On page 4, the authors discuss the accuracies of these prediction methods. I think for the readers who are less familiar with these tools, a brief discussion of what this accuracy means - or how the authors of the tools intend for accuracy to be interpreted - would be useful. This accuracy is especially surprising given the statement on page 4, line 47-48, citing how in silico assessments are at odds with in vitro evidence. Again, this is not surprising to those who are experts in this field, but for the general readership, a bit more explanation would be helpful.

Response: Five of the methods we evaluated only reported AUROC values, rather than traditional accuracy metrics (only GWAVA does). Thus, the published scores lead to somewhat subjective interpretation. We have now added explanations of the predictive accuracies of these methods in the revised manuscript and caution the users about the practice of relying on AUROC.

Comment R2.c. On page 6, lines 85-88, it is a bit surprising that there is a complete overlap of pathogenicity scores for pathogenic and non-pathogenic ncSNVs; isn't this what these methods are intended to do? Differentiate between pathogenic and non-pathogenic?

Response: Indeed, all the methods claim to distinguish pathogenic from non-pathogenic variants, and that is how they are used by practitioners. The complete overlap between impact scores of pathogenic and non-pathogenic alleles in task 1 is because these alleles share the same position while the prediction models are trained using variants at widely separated positions. Thus, the failure to perform well on task #1 means that these methods are not designed to capture the characteristics of different alleles at the same position. We have now added the explanation in the Discussion section of the revised manuscript.

Comment R2.d. On page 7, the authors mention the use of the HGMD database for selection of the variants. I also looked in the methods on this topic. Was anything done to prohibit the number of false positive variants from HGMD - as HGMD is known to have a large number of false positive variants in the database. Was anything done in the selection of variants to minimize the number selected?

Response: We were mindful of this problem and had applied two filters to remove potential false positives in the HGMD database in the original manuscript. These two filters corresponded to two evidences of pathogenicity in the clinical guidelines published by the ACMG (PMID: 25741868). Specifically, the first filter removed HGMD variants located outside known regulatory elements (evidence PM1 in the ACMG guidelines). The second filter removed HGMD variants observed in the 1000 Genome Project populations with >1% frequencies (evidence PM2 in the ACMG guidelines). We have now added a third filter that removed HGMD variants labelled as "DM?", which indicates a degree of uncertainty, and those labelled as "DP," which indicates disease-associated variants but with no functional evidence (evidence PS3 in the ACMG guidelines). Considering that the HGMD database collects DAVs from published studies showing segregation between cases and controls (corresponding to the evidence PS4), our filtering criteria meet or exceed the requirements to diagnose pathogenic variants in the ACGM guidelines. The results obtained with two versus three filters were very similar, even though the latter collection is smaller (count of HGMD DAVs = 2,037 with no filter, 957 with two filters, and 764 with three filters). We have described these three filters in details in the Methods section and report the results using three filters in the manuscript and the results using two filters in the supplementary materials (see Supplementary Figures 3 and 6).

Comment R2.e. On page 15, lines 244-246, the authors discuss that each eQTL effect could be the summation of minor contributions from multiple variants within an interval. I think that this is an interesting and likely reasonable explanation. I wonder if there is a way to test this theory in your simulations? Or perhaps expanding on this idea bit would be useful. I think as more people consider new pathogenicity prediction algorithms.

Response: We appreciate this comment. However, the test of this theory will be complicated and requires a thorough investigation. At present, without comprehensive data supporting this theory, we refrained from expanding our discussion on this hypothesis in the present manuscript.

Comment R2.f. All in all, I think that this is a really important contribution to the literature. Too many researchers are using one of these types of tools and moving research forward based on these predictions, which we are learning are not very accurate. This series of analyses are providing a nice framework from which to move the field forward to develop better algorithms.

Response: Thank you. We agree that our analyses will be very useful for a large scientific community, who are using the tools we have evaluated for the biological tasks we have outlined.

Reviewer #3

Comment R3.a. In “Biological relevance of computationally predicted pathogenicity of noncoding variants”, Liu et al assess six commonly used variant annotation tools in terms of their ability to differentiate sets of non-coding variants with or without disease associations. They conclude that these tools have minimal power to differentiate pathogenic from non-pathogenic variants within regulatory regions. I find that there are a few valuable points and interesting ideas in this manuscript, the general topic is an important one, and there is little doubt that new and improved annotations are needed to better discriminate phenotypically relevant non-coding variants.

Response: Indeed, this is an important topic in functional genomics, as these six methods are being used extensively in research for biological tasks we have presented. The main message of our manuscript is cautionary, which is about biology and methodology. It may be useful to note that all these six methods have been published in high profile journals (four in *Nature Genetics* and two in *Nature Methods*), all in the past four years, surely highlighting the importance of the research problem being addressed. Also, these articles have already been collectively cited over 2,000 times in a short span of time. These facts coupled with our many conversations with colleagues indicate an increasing level of frustration with application of the methods and much interest in our findings given the implications for how they are used in ongoing research. Therefore, the topic is of exceptional importance and interest.

Comment R3.b. However, the core conclusions rest upon unrealistic, or at best unproven, assumptions. Details are below, but the high-level summary is that the criteria used to define the “pathogenic” and “non-pathogenic” test sets are error-prone, and insufficient effort is made to account for these test set errors at either the bulk or individual variant level. These test set errors are likely to be large, but are at the very least not negligible, and would tend to make the annotations appear to perform more poorly than they actually do.

Response: We request the reviewer to reconsider this sentiment, because our assumptions are the same as those made in the original research articles describing the six methods we evaluated. We have followed well-established traditions in the field to define “pathogenic” and “non-pathogenic” for all the biological tasks. We have clearly explained these facts in responses *R3.d* and *R3.i* below. Nevertheless, we have followed the suggestions and performed additional analyses on a new data source (i.e., ClinVar, see Supplementary Figures 4 and 7) and on more stringently filtered HGMD DAVs and evolutionary alleles. We have ensured that data set errors are expected to be lower than those in the most prominent articles in the field. Results from these new analyses are consistent with those in the original submission.

Comment R3.c. As a result of these issues, and some other more minor ones, I do not find the conclusions drawn in the manuscript to be well supported. In fact, some of the results suggest the opposite conclusion. My more detailed comments will be structured within the three “biological tasks” that are used in the manuscript.

Response: We have provided solid rationale for the test data selection and carried out reanalysis of these data under more stringent criteria. The new results are similar to those presented in the original submission, establishing the robustness of our conclusions to the concerns raised by R3. We also show that the statement that “some of the results suggest the opposite conclusion” is unwarranted (see response *R3.d* below).

Comment R3.d. The first task studied is the ability for annotations to differentiate distinct alleles at a given genomic position. To do so, a set of “non-pathogenic” alleles is defined as those with a minor allele frequency between 5% and 15% that are absent from GWAS catalogs. A set of “pathogenic” alleles is then defined as those alleles at the same genomic position that are not observed in non-human primates, as these are “presumably deleterious (non-neutral)”. *This presumption is essential to the analysis but not justified or critically examined.* It is also the opposite of the null model, which is an assumption of neutrality. This is particularly true for SNV alleles at a genomic position that is known to carry at least one high-frequency allele and which therefore have a higher than background probability of being neutral. In light of this null, substantive evidence is required to infer that any given allele or set of alleles is not neutral. Towards that end, the analysis presented relies upon the absence of an allele from an unspecified number of primate assemblies that span an unquantified amount of evolutionary divergence. However, no estimation is provided as to how well such evidence actually differentiates deleterious from neutral alleles, especially among alleles where the prior probability of neutrality is particularly high. As an extreme example, consider a scenario wherein only the chimpanzee reference assembly is used; the absence of an alternative allele in this assembly has very little information content and nearly every possible allele would be labeled as “presumably deleterious”. Obviously, inclusion of more primates improves the signal, but even the complete repertoire of available primate assemblies is unlikely to have a high positive predictive value for inferring pathogenicity in this case. My belief is that the observed lack of separation by any of the annotations between the two sets of variants in this task is overwhelmingly due to a genuine lack of biological separation.

Response: To begin with, for four of the methods (LINSIGHT, EIGEN, GWAVA, and CATO), our conclusions regarding task #1 are robust to any difficulties in assembling the test sets, because these methods produce identical scores for all non-reference alleles at the same position. We regret that this point was not made clearly in the initial submission.

Second, for the other two methods (CADD and DeepSEA), our tests sets are well justified and follow the convention in the field. Our definition of pathogenic alleles is in line with the training data used in CADD development. Specifically, CADD uses de novo mutations to represent pathogenic variants, as they select bases that are different from the human-chimpanzee ancestral alleles and different from the human reference allele, presumably under purifying selection. We have used a more strict definition by considering evolutionary history beyond human-chimpanzee and used 56 diverse mammals (see the Methods section). Also, our definition of non-pathogenic alleles is shared by five of the six methods (CATO as the only exception) that consider mutations with significant population frequency or presence among species are likely to be neutral under the prevailing theory of molecular evolution. Given these common backgrounds, the uniqueness of our test set in task 1 is to require pathogenic and non-pathogenic alleles share the same position. In our view, the evaluation of the performance of six methods for this task is very meaningful, as the biological and evolutionary properties of positions are the same in this balanced test set, which allows us to investigate the allelic effects. *For these reasons, we consider the demand of “substantive evidence ... to infer that any given allele or set of alleles is not neutral,” to be an overly stringent burden of proof, which was not placed on any of the previous studies which used de novo mutations without any filtering to be pathogenic.*

Thirdly, we have extended the evolutionary history by expanding the set of non-human species to 56 placental mammals, which resulted in an evolutionary time span of ~2.9 billion years (sum of branch lengths over the whole phylogenetic tree, see the Methods section). Given the mutation rate of the order of 10^{-8} - 10^{-9} per base per year, the use of so many placental mammals ensures that each position has mutated many times and that the un-observed bases at each selected position have been purified due to deleteriousness, all of these un-observed SNVs are expected to have emerged but been purified due to deleteriousness. Using this phylogeny, which encompasses 10-times longer evolutionary time than the previous collection of 11 primate species, we obtain the same result (Figure 1) as that reported in the original manuscript (see Supplementary Figure 1B). So our results are robust to the diversity of the species used.

In any case, our conclusions about task #1 are robust to the reviewer's concerns.

Comment R3.f. Why 5-15% MAFs to select “neutral” variants? What about other parameter choices? Why impose a maximum MAF?

Response: The 5%-15% MAF was imposed in task #1 to select the putatively neutral variant sharing the same genomic position with a deleterious variant. Within this MAF range, we intend to eliminate variants that are potentially under positive selection or balancing selection. In any case, we have also performed the analysis with no restriction on MAF, and the results did not change (see Supplementary Figure 1A).

Comment R3.g. To what extent are the ncSNVs that are presumed neutral as a result of not being detected by a GWAS likely to include genuinely disease-associated variants that have simply not yet been detected? This is the converse to my main concern, and admittedly likely to have a smaller impact, but is another potential artifact that would narrow any genuine biological gap between the two variant sets.

Response: We are aware of the false negatives due to the low power of GWAS studies. Therefore, when we used the GRASP database as a filter to remove potentially disease-associated variants from the population polymorphisms, we used a lenient p-value cutoff of 0.05. Although this may still leave some false negative variants in the control samples, we expect the percentage of such variants to be highly reduced. No method is perfect, but we feel this is also consistent with the published best practices.

Comment R3.h. Many details on the selection of “presumably deleterious” alleles are missing. Which primates were analyzed? Were all ncSNVs tested, or only those at positions that were aligned to other primates? If not all primates were required to be aligned at that position, how many were required? Were primate population genetic datasets examined to look for within-species polymorphisms, or just the reference assemblies? Etc.

Response: We have now provided all of these details in the Methods section and noted that we primarily used the reference assemblies and that less than 1% of the presumably deleterious alleles occur as polymorphisms in the Great Ape Genome Project data and were removed.

Comment R3.i. The second biological task is more credible than the first, but only mildly so. The basic concern in task one – i.e., that the variant set labeled as “pathogenic” is contaminated with a large but unestimated proportion of actually non-pathogenic variants – also holds for task two. In this case, pathogenicity is defined by presence in HGMD coupled to the existence of “demonstrated effects ... on in vitro transcription”. The latter assertion is not explained or qualified; i.e., what standards of regulatory effect size, statistical support, assay type, etc., were used to define “demonstrated effects”? More important is the issue of pathogenicity, which is more difficult to assess and for which evidence of transcriptional effect is neither necessary nor sufficient. The conclusion that these DAVs are actually pathogenic depends on the presumption that presence in HGMD equates to being pathogenic. However, this is not true, as the false positive rate of HGMD is substantial (a long of the comment with many citations to the literature).

Response: We are fully aware of the literature summarized by R3, and we appreciate his/her attention to this important detail. This was the reason why we had already filtered HGMD variants from original 2,037

DAVs to 957 DAVs. In the revised version, we have further filtered them down to 764 DAVs using criteria that meet or exceed the recommendations in the ACMG guidelines. We described these filters in the revised manuscripts and in the response *R2.d* above. In addition, we have followed R3 advice and analyzed the ClinVar dataset. This analysis yielded patterns similar to those for HGMD variant sets (see Supplementary Figures 4 and 7). Overall, we have taken many steps to exclude potentially non-pathogenic variants, which shall result in an enrichment of pathogenic variants in the final sets of filtered HGMD as well as ClinVar variants analyzed. For these reasons, the second task of fine mapping, which is actually a mainstay of functional genomics research, is highly credible.

Comment R3.j. I find it likely that a large fraction of the “pathogenic” DAVs in task two are actually non-pathogenic and, as for task one, the results presented thus fail to separate predictor error from a genuine lack of biological differentiation between “pathogenic” and “non-pathogenic”. And, again, as for task one even if I am wrong it is incumbent on the authors to justify their definitions and provide evidence that they accurately separate truly pathogenic and truly non-pathogenic alleles.

Response: We have fully presented our definitions, and made additional efforts to reduce impurities in the dataset of pathogenic variants (see our response *R3.i* above). We have been as conservative as possible, given the nature of the state-of-the-art data available today. Our choices are well-justified for task #2, because we have followed current standards in the field – indeed, the GWAVA, LINSIGHT and DeepSEA, each include an evaluation of their tool based on a causal set including all the HGMD regulatory variants and a control set consisting of variants from the 1kG resource, while our evaluation relies on a much more stringently filtered set of HGMD variants.

Comment R3.k. I note that the above concern is acknowledged in the manuscript but ignored without explanation. Specifically, the text beginning at line 127 explains that variants disrupting conserved promoters are more accurately distinguished by prediction methods and that such variants “may also be the most definitively defined in the HGMD database.” Assuming that “most definitively defined in the HGMD database” means most likely to truly be pathogenic, I agree. The fact that the predictors more effectively separate conserved promoter DAVs from CPPs is, in my opinion, a direct result of the fact that conserved promoter DAVs in reality have a lower false positive rate with respect to disease relevance. In this scenario the predictors are producing the correct outcome, a result that contradicts the main conclusions of the manuscript.

Response: This concern is fully addressed by the fact that we have now cautiously applied three filters on the HGMD variants (see our response *R3.i* above). In addition, we have also validated results from HGMD analysis by conducting an analysis of ClinVar variants, as noted above. The conclusions are no different from those reported in the original manuscript.

Comment R3.l. There is a lot of missing information about the set of HGMD variants. What processing was applied to HGMD? What does the resulting collection of variants look like in terms of GnomAD/ExAC allele frequencies, distance to TSSs, distances to exon junctions, conservation, etc.?

Response: Indeed, some of this basic information will be useful for the readers, so we have added it in the revised manuscript (see the Methods section). In particular, we have noted that only 32 out of 764 had a population frequency greater than 1% in gnomAD, 515 were within 1kb of a transcription start site (because of our requirement of DAVs in known regulatory elements, which are enriched in proximal regions of TSS), 19/45 intronic variants were within 1kb of an exon junction, and 239/335/190 were at ultra-/well-/least-conserved sites, respectively.

Comment R3.m. How many genes/loci are represented and how sensitive are the results to the fact that large fractions of HGMD alleles cluster near a small number of genes?

Response: Following R3 suggestion, we examined the distribution of HGMD DAVs among genes and found they are clustered unevenly around 318 genes. To investigate the effect of the clustering patterns on the performance evaluation, we randomly chose one DAV in the flanking region of a gene and constructed

a gene-balanced test set. Compared to the analyses using all the HGMD DAVs, we observed slightly worse performance on the gene-balanced test set, which have been included in the revised manuscript (Supplementary Figures 5 and 8).

Comment R3.n. Why use HGMD data from 2015 rather than 2018?

Response: We used HGMD data from 2015, because we had purchased the professional version of the database at that time (US \$5,000). We did not purchase the newest HGMD 2018 release, because (a) the set of regulatory HGMD 2015 variants is anticipated to contain a vast majority of the HGMD 2018 collection of regulatory variants, (b) the results for the stringently filtered HGMD 2015 version and the leniently filtered HGMD 2015 version are concordant, and (c) results from the analysis of ClinVar data are similar to HGMD 2015 results (see response R3.i). Therefore, we feel that the HGMD 2015 and ClinVar collections are sufficient to generate reliable patterns.

Comment R3.o. Why not use ClinVar, a free, open-access, community-driven tool that provides more and better information pertaining to the pathogenicity of any given variant, including how many and which labs scored the variant, the standards used by the reporting labs, the scores assigned by each lab, etc.?

Response: We have now done that. We found 272 ClinVariants noncoding DAVs that are absent from the 1000 Genomes Project (see the Methods section) and used them in tasks #2 and #3. We found that the results are very similar to those obtained using HGMD 2015 variants. We have now mentioned these result in the main text (see Supplementary Figures 4 and 7).

Comment R3.p. The LD-based analyses on page 8 are not clear as written. I believe the r-sq thresholds are used to select the blocks of CPPs near to a DAV, regardless of the correlation between the DAVs and the CPPs per se. As written the suggestion is that the CPPs are selected as a result of pairwise correlations with the DAVs, which I do not believe is the case.

Response: We have now rephrased the sentence to avoid the confusion.

Comment R3.q. Task three depends on variant definitions from task two, and therefore suffers from the same problems.

Response: This problem about variant definitions has been addressed above, and we obtained similar results from different datasets in task 3 (see Supplementary Figures 6-8). Our motivation behind test #3 is to evaluate the dependence of appropriate accuracy measures on composition of the data set being evaluated, which follows directly from the theory of positive and negative predictive values. No matter how good the predictor is, if the mixing ratio is low, precision will drop. This is a critical point that is not addressed in articles describing widely used methods.

Reviewer #1 (Remarks to the Author):

The authors have responded appropriately to the points raised. No additional comments from this reviewer.

Reviewer #2 (Remarks to the Author):

The authors have done a brilliant job in their revision. I appreciate the level of attention spent on all of the previous reviewer comments. Their analyses are thorough and the interpretations are sound.

My only additional suggestion is that it could be even more impactful and beneficial for the non-expert reader if you add a summary table to the discussion that provides an overview of where each method has any utility. You could list the methods on one axis and the different scenarios that you tested on the other, with check boxes where the method performs well. I recognize that most boxes of such a table would go unchecked and certainly across the methods, there is little consistency. If there is a way in one image to summarize all of this impressive work, I think it could make more of the community realize how nontrivial these nsSNV annotations are and how little emphasis we should place on the pathogenic ones since the labels are so inconsistent.

Reviewer #3 (Remarks to the Author):

The major concerns raised in my initial review hold true in my evaluation of the revised manuscript. I again agree that this is an important area where new methods and approaches are needed. However, this study does not present results or conclusions that I find to be credible or informative. There remains insufficient effort to critically examine the assumptions being made and quantify the true and false positive rates among the "pathogenic" sets of variants, despite being essential to evaluate the conclusions drawn. I do not agree that "well-established traditions" are being followed. If other studies depended on unobserved alternative alleles at sites of high-frequency variation to be deleterious, or that HGMD pathogenicity assertions are highly accurate, then those studies deserve the same type of scrutiny and skepticism.

Task 1

While more information is now provided as to how "evolutionarily forbidden" alleles were defined, I still disagree with the assumption that because a given allele has not been observed to be fixed in one or more of the compared mammals that the given allele must be deleterious. The presumption of deleteriousness is converse to what should be the default presumption, i.e., neutrality, as most positions in human genomes are mutable with little or no fitness consequence. Further, for a given variant genomic position, the fact that it harbors one common allelic variant increases the probability that unobserved alleles at that site would also be neutral.

The preceding factors together necessitate a high bar to conclude that the as-defined "evolutionarily forbidden" alleles are truly deleterious. Towards that end, the revised text points to the fact that the 56 compared mammalian genome assemblies span 2.9 billion years of evolution and that mutation rates are 10^{-8} per base per year. I would certainly agree that these factors assure that any given allele has arisen many times as a de novo event in an individual animal these populations. However, the de novo mutation rate is not being examined in this analysis and is not the relevant quantity; rather, by virtue of comparing reference genome assemblies across these species, what is being examined are mostly fixed alleles ("substitutions"), and thus the site-specific substitution rate is the critical quantity. In turn, in order for an observation of zero substitution events of a given allele to be meaningfully indicative of deleteriousness, the mean expectation of the number of substitution of that allele must be relatively high. While no estimates of neutral substitution rates are provided, my guess is that it is likely to be at most (i.e., in sites where the alignment includes all 56 mammals) around six substitutions per site (Lindblad-Toh et al (2011)). With three possible alternative alleles, and an observed rate of six substitutions per site, the expected average number of observed fixation events for a particular allele would be two;

thus, zero will frequently occur at random without any influence of purifying selection. The fact that many sites will not be aligned across all 56 mammals (and thus, fewer than 6 substitutions per site are being assayed for these positions) and that not all neutral alleles have equal rates of fixation owing to unequal mutation rates (e.g., transitions vs transversions), would further increase the rate at which many truly neutral alleles would have an observed fixation rate of zero.

Thus, the presumption of deleteriousness is not only converse to the appropriate null model but is also converse to the data. I thus continue to believe that the lack of separation between alleles in task 1 is an accurate reflection of the fact that both categories of alleles are overwhelmingly truly neutral.

Disease-associated variant (DAV) analysis

While the efforts to explain, clean up, and compare with ClinVar are valuable, I believe the conclusions drawn still depend on an insufficiently skeptical view of the true rates of pathogenicity within these variants. As an initial matter, observing that 32 of 764 DAVs have GnomAD frequencies greater than 1% is not encouraging, as this indicates that at least 4.2% of the retained variants are not highly penetrant contributors to disease. The fact that 4% is a minimal false positive rate certainly does not inspire confidence that the remaining 96% are reliably pathogenic. Additional reasons to doubt the reliability of the DAV analysis are as follows.

First, the revised analysis claims to use ACMG guidelines for determining pathogenicity. However, multiple ACMG evidence codes are assigned to variants based on HGMD-reported assessments, rather than evaluation of the actual data for each variant. For example, PS3 and PS4 are assumed to be applicable to variants based on the presence/absence of HGMD codes like "DM?" or "DP". Such indirect assignment of evidentiary support is not allowed under ACMG guidelines; PS3 and PS4 (and most other codes) can only be used when one has evaluated the actual data for that variant and found them to support the use of those codes. To the extent that one relies on an external evaluation of the actual data, PP5 ("supporting evidence") should be used - i.e., "Reputable source recently reports variant as pathogenic, but the evidence is not available to the laboratory to perform an independent evaluation". Further, to use PP5 the source report needs to be "recent", a critical feature given that older pathogenicity assertions are especially unreliable; no mention of date ranges of HGMD assertions, besides being before 2015, are provided, making it unclear to what extent even PP5 can be used.

Second, beyond the concern that the variant evaluations are not being performed with the rigor required under ACMG rules, the more fundamental problem in this study is that HGMD's evidentiary assertions are given the benefit of the doubt, with filters set up only to remove obvious errors in these assertions. However, many HGMD pathogenicity assertions are based on flimsy evidence, with typical examples including absence of an allele from 96 control samples, perfect "segregation" as a result of observing transmission from an affected parent to a single affected child, or functional assays with nebulous molecular (let alone clinical) results. The net result of these problems, and which in my previous comments I supplied references to support, is that HGMD statements cannot by themselves be assumed to reliably indicate pathogenicity (it is for precisely this reason that ACMG rules place only minimal weight on database-level representation of pathogenicity, and place no weight on older such representations). Robust, proactively supportive evidence for the pathogenicity of individual variants is required.

Third, using 1000 Genomes frequencies as a filter rather than GnomAD/ExAC frequencies further inflates error rates. GnomAD/ExAC frequency estimates are far more accurate, especially for lower-frequency variation where small differences in absolute frequency matter greatly to the interpretation. Related to this point, a 1% allele frequency cutoff is, for nearly all Mendelian diseases, orders of magnitude too high to be effective. While in principle allele frequencies must be interpreted relative to the expected prevalence of the disease being considered, typical cutoffs are far lower, on the order of 0.1% (which is itself too high for most known rare disorders). See Whiffin et al (2017) for a thorough assessment of this topic.

Fourth, while being below a given frequency cutoff is sufficient to avoid invoking BS1 or BS2 (strong support that a variant is benign based on allele frequencies greater than expected given

disorder prevalence), it is not sufficient to invoke the moderate pathogenicity evidence code PM2, as is done in this study. PM2 requires that a variant is "absent from controls (or at extremely low frequency if recessive)". While GnomAD/ExAC are not strictly speaking "controls", they are not ascertained for any particular rare disease or phenotype, and severe pediatric disease individuals have been removed. Thus, for most diseases PM2 is applied for variants that are completely absent from GnomAD/ExAC, while for more highly prevalent conditions PM2 might be justified if seen in handfuls of individuals. Use of PM2 is generally questionable for alleles seen dozens to hundreds of times in GnomAD/ExAC (allele frequencies between 0.01% and 0.1%), and is clearly problematic for variants seen in hundreds to thousands of individuals (0.1% to 1%).

Finally, the text states that by removing variants "outside known regulatory elements", PM1 becomes appropriate to use. However, this is an overly liberal usage of PM1, which states that a variant is in a "critical and well-established functional domain (e.g., active site of an enzyme) without benign variation". The text describing PM1 in the ACMG guidelines paper further describes it as applying to "certain protein domains" that are "critical to protein function" and in which "all missense variants identified to date in these domains have been shown to be pathogenic". It is debatable whether any regulatory element qualifies for use of this code, but even if there exist individual regulatory elements for which this code is appropriate, merely being inside "a regulatory element" would not be sufficient to invoke it, much as merely being inside "a protein domain" is also insufficient.

In general, I believe this study falls well short of providing sufficient evidence that the DAVs as defined are reliably pathogenic. The concerns described above would be relevant to all claims of pathogenicity, but rigor is particularly important for variants that do not disrupt proteins and thus have intrinsically higher a priori probabilities of not being truly disease-causal. Such standards may be inconvenient (and clearly we all wish there was a large collection of truly highly penetrant non-coding variants), but cannot simply be ignored when drawing conclusions that depend on reliable definitions of pathogenicity.

Specific Responses to Reviewers' Comments and Suggestions

We thank all the reviewers for their constructive comments. We are pleased to see that two reviewers (R1 and R2) were satisfied with our responses and approved our revision. R3 still has reservations, which we address in this revision. In particular, we have revised the manuscript following three specific suggestions made by the handling editor, which we respond to first below. This is followed by detailed explanation of how we considered all the other comments by the reviewers.

Response to Editor's comments

(EC#1) Editor's Comment #1: Use GnomAD/ExAC to filter HGMD variants

Response: We have further reanalyzed the data with two thresholds for GnomAD frequency: one suggested by R3 (0.01%) and an even more stringent criterion (0%). These analyses produce patterns not meaningfully different from those obtained using the 1KG filter and showed that there is little relationship between population frequencies of rare pathogenic variants and the predicted pathogenicity scores. We have now reported results also using gnomAD in the revised manuscript (Supplementary Figs. 3, 4, 5 and 8).

(EC#2) Editor's Comment #2: Definition of pathogenic variants, full compliance with ACMG.

Response: ACMG has NO guidelines for defining pathogenic variants in the noncoding regions; these guidelines apply only to coding variants. In order to be as thorough and conservative as possible, we applied ACMG guidelines as a best practice whenever feasible to use for noncoding variants. That is, we set new standards, beyond those followed by authors of the six methods analyzed. Unfortunately, further compliance with the remaining ACMG guidelines would require manual data curation from thousands of publications, which is beyond the scope of the current manuscript. Because similar results are seen for both HGMD and ClinVar variants, we feel that all the feasible analyses point to the same patterns, particularly in light of the absence of dependence on MAF as noted in EC#1 above. Per editorial suggestion, we have now included a supplementary table containing all the pathogenic variants after applying the filters (Supplementary Table 3).

(EC#3) Comment #3: Reconsideration of the definition of "evolutionarily forbidden alleles"

Response: This comment applies to the test dataset for only the first task. For this task, we begin by noting that four out of six methods produced the same score for all the alternative alleles at every position tested. Therefore, our conclusions for those four methods (CATO, EIGEN, GWAVA, and LINSIGHT) are not affected by R3's concern about evolutionary forbidden alleles. For the other two methods (CADD and DeepSEA), we can justify the use of "evolutionarily forbidden" alleles from three aspects: (1) Molecular evolutionary principles have been used in the development of CADD and LINSIGHT, two of the methods tested here, so these principles are appropriate to use. (2) Existing literature on comparative genomics has shown that "*the probability that a genomic sequence is not under purifying selection will remain fixed across all 29 species is < 0.02 for single bases*" (Lindblad-Toh et. al., 2011). With a larger set of species (58), this probability becomes even smaller in our simulations (<0.006, see the Online Methods section). We now show that the results remained unchanged even when we sampled pathogenic alleles only from positions that were completely conserved across all 58 mammalian species analyzed and contained no alignment gaps (AUROC = 0.49 and 0.57 for CADD and DeepSEA, respectively; see the Online Methods section).

Response to Reviewers' comments

Reviewer #1

Comment R1. The authors have responded appropriately to the points raised. No additional comments from this reviewer.

Response: Thank you.

Reviewer #2

Comment R2.a. The authors have done a brilliant job in their revision. I appreciate the level of attention spent on all of the previous reviewer comments. Their analyses are thorough and the interpretations are sound.

Response: Thank you. Your suggestions have also helped us improve the quality of this manuscript.

Comment R2.b. My only additional suggestion is that it could be even more impactful and beneficial for the non-expert reader if you add a summary table to the discussion that provides an overview of where each method has any utility. You could list the methods on one axis and the different scenarios that you tested on the other, with check boxes where the method performs well. I recognize that most boxes of such a table would go unchecked and certainly across the methods, there is little consistency. If there is a way in one image to summarize all of this impressive work, I think it could make more of the community realize how nontrivial these nsSNV annotations are and how little emphasis we should place on the pathogenic ones since the labels are so inconsistent.

Response: We have now added Table 2 that summarizes and compares the utilities of the six methods.

Reviewer #3

Comment R3.a. The major concerns raised in my initial review hold true in my evaluation of the revised manuscript. I again agree that this is an important area where new methods and approaches are needed. However, this study does not present results or conclusions that I find to be credible or informative. There remains insufficient effort to critically examine the assumptions being made and quantify the true and false positive rates among the “pathogenic” sets of variants, despite being essential to evaluate the conclusions drawn. I do not agree that “well-established traditions” are being followed. If other studies depended on unobserved alternative alleles at sites of high-frequency variation to be deleterious, or that HGMD pathogenicity assertions are highly accurate, then those studies deserve the same type of scrutiny and skepticism.

Response: Our conclusions and the *Discussion* section clearly reflects the sentiment that all previous methods, their approaches, and their claims need to be scrutinized. In fact, our manuscript represents one such scrutiny. As for concerns about the evolutionarily forbidden alleles, we have now presented a more detailed response (see EC#3). Also, we did not assume that HGMD pathogenicity assertions are highly accurate, but rather applied very stringent filters to remove potential false positives (see EC#2). Therefore, we not only followed the well-established traditions, but have advanced significantly beyond them in our analyses to ensure accurate assertions (see EC#1 and EC#2).

Comment R3.b. Task 1: While more information is now provided as to how “evolutionarily forbidden” alleles were defined, I still disagree with the assumption that because a given allele has not been observed to be fixed in one or more of the compared mammals that the given allele must be deleterious. The

presumption of deleteriousness is converse to what should be the default presumption, i.e., neutrality, as most positions in human genomes are mutable with little or no fitness consequence. Further, for a given variant genomic position, the fact that it harbors one common allelic variant increases the probability that unobserved alleles at that site would also be neutral.

Response: This concern is fully addressed; see our response to EC#3.

Comment R3.c. The preceding factors together necessitate a high bar to conclude that the as-defined “evolutionarily forbidden” alleles are truly deleterious. Towards that end, the revised text points to the fact that the 56 compared mammalian genome assemblies span 2.9 billion years of evolution and that mutation rates are 10-8 per base per year. I would certainly agree that these factors assure that any given allele has arisen many times as a de novo event in an individual animal these populations. However, the de novo mutation rate is not being examined in this analysis and is not the relevant quantity; rather, by virtue of comparing reference genome assemblies across these species, what is being examined are mostly fixed alleles (“substitutions”), and thus the site-specific substitution rate is the critical quantity. In turn, in order for an observation of zero substitution events of a given allele to be meaningfully indicative of deleteriousness, the mean expectation of the number of substitution of that allele must be relatively high. While no estimates of neutral substitution rates are provided, my guess is that it is likely to be at most (i.e., in sites where the alignment includes all 56 mammals) around six substitutions per site (Lindblad-Toh et al (2011)). With three possible alternative alleles, and an observed rate of six substitutions per site, the expected average number of observed fixation events for a particular allele would be two; thus, zero will frequently occur at random without any influence of purifying selection. The fact that many sites will not be aligned across all 56 mammals (and thus, fewer than 6 substitutions per site are being assayed for these positions) and that not all neutral alleles have equal rates of fixation owing to unequal mutation rates (e.g., transitions vs transversions), would further increase the rate at which many truly neutral alleles would have an observed fixation rate of zero.

Response: We appreciate R3 insight and conducted additional analysis, which yielded results similar to those reported before. Basically, we compiled a highly restricted subset of positions in which pathogenic alleles were sampled only from positions that were completely conserved across all 58 species analyzed and did not contain any alignment gaps. This maximized our chances of sampling pathogenic alleles, because completely conserved positions are rare when species sampling is diverse and evolution is strictly neutral. For example, Lindblad-Toh et al. estimated it to be less than 2% for a set of 29 mammals. In our simulations, we found that the completely conserved positions are expected to occur even more rarely in our datasets (< 0.6%); we used a 58 placental mammal species timetree, a realistic neutral evolutionary substitution rate, a high transition-transversion rate bias, and an extreme range of G+C content biases (see the Online Methods section as well as our response to EC#3 above).

Comment R3.d. Thus, the presumption of deleteriousness is not only converse to the appropriate null model but is also converse to the data. I thus continue to believe that the lack of separation between alleles in task 1 is an accurate reflection of the fact that both categories of alleles are overwhelmingly truly neutral.

Response: For task 1, we note that four out of six methods produced the same score for all the alternative alleles at every position tested. Therefore, our conclusions for task 1 is valid for four methods (CATO, EIGEN, GWAVA and LINSIGHT) irrespective of R3’s concern. Our responses in EC#3 and R3.c provide justification for the use of “evolutionarily forbidden” alleles.

Comment R3.e. Disease-associated variant (DAV) analysis. While the efforts to explain, clean up, and compare with ClinVar are valuable, I believe the conclusions drawn still depend on an insufficiently skeptical view of the true rates of pathogenicity within these variants. As an initial matter, observing that 32 of 764 DAVs have GnomAD frequencies greater than 1% is not encouraging, as this indicates that at least 4.2% of the retained variants are not highly penetrant contributors to disease. The fact that 4% is a

minimal false positive rate certainly does not inspire confidence that the remaining 96% are reliably pathogenic. Additional reasons to doubt the reliability of the DAV analysis are as follows.

Response: DAVs with >1% population frequency are not necessarily false positives. Nevertheless, we followed R3 suggestion and found that the result was not different. This is explained further in response to EC#1.

Comment R3.f. First, the revised analysis claims to use ACMG guidelines for determining pathogenicity. However, multiple ACMG evidence codes are assigned to variants based on HGMD-reported assessments, rather than evaluation of the actual data for each variant. For example, PS3 and PS4 are assumed to be applicable to variants based on the presence/absence of HGMD codes like “DM?” or “DP”. Such indirect assignment of evidentiary support is not allowed under ACMG guidelines; PS3 and PS4 (and most other codes) can only be used when one has evaluated the actual data for that variant and found them to support the use of those codes. To the extent that one relies on an external evaluation of the actual data, PP5 (“supporting evidence”) should be used - i.e., “Reputable source recently reports variant as pathogenic, but the evidence is not available to the laboratory to perform an independent evaluation”. Further, to use PP5 the source report needs to be “recent”, a critical feature given that older pathogenicity assertions are especially unreliable; no mention of date ranges of HGMD assertions, besides being before 2015, are provided, making it unclear to what extent even PP5 can be used.

Response: This concern is fully addressed; see our response to EC#2.

Comment R3.g. Second, beyond the concern that the variant evaluations are not being performed with the rigor required under ACMG rules, the more fundamental problem in this study is that HGMD’s evidentiary assertions are given the benefit of the doubt, with filters set up only to remove obvious errors in these assertions. However, many HGMD pathogenicity assertions are based on flimsy evidence, with typical examples including absence of an allele from 96 control samples, perfect “segregation” as a result of observing transmission from an affected parent to a single affected child, or functional assays with nebulous molecular (let alone clinical) results. The net result of these problems, and which in my previous comments I supplied references to support, is that HGMD statements cannot by themselves be assumed to reliably indicate pathogenicity (it is for precisely this reason that ACMG rules place only minimal weight on database-level representation of pathogenicity, and place no weight on older such representations). Robust, proactively supportive evidence for the pathogenicity of individual variants is required.

Response: Our analysis using ClinVar data (suggested by R3) produced very similar patterns to what was found using HGMD data. So, our conclusions are robust (see also response to EC#2).

Comment R3.h. Third, using 1000 Genomes frequencies as a filter rather than GnomAD/ExAC frequencies further inflates error rates. GnomAD/ExAC frequency estimates are far more accurate, especially for lower-frequency variation where small differences in absolute frequency matter greatly to the interpretation. Related to this point, a 1% allele frequency cutoff is, for nearly all Mendelian diseases, orders of magnitude too high to be effective. While in principle allele frequencies must be interpreted relative to the expected prevalence of the disease being considered, typical cutoffs are far lower, on the order of 0.1% (which is itself too high for most known rare disorders). See Whiffin et al (2017) for a thorough assessment of this topic.

Response: We have now used GnomAD/ExAC, and the result did not change, as noted in the response to EC#1.

Comment R3.i. Fourth, while being below a given frequency cutoff is sufficient to avoid invoking BS1 or BS2 (strong support that a variant is benign based on allele frequencies greater than expected given disorder prevalence), it is not sufficient to invoke the moderate pathogenicity evidence code PM2, as is done in this study. PM2 requires that a variant is “absent from controls (or at extremely low frequency if recessive)”. While GnomAD/ExAC are not strictly speaking “controls”, they are not ascertained for any particular rare disease or phenotype, and severe pediatric disease individuals have been removed. Thus,

for most diseases PM2 is applied for variants that are completely absent from GnomAD/ExAC, while for more highly prevalent conditions PM2 might be justified if seen in handfuls of individuals. Use of PM2 is generally questionable for alleles seen dozens to hundreds of times in GnomAD/ExAC (allele frequencies between 0.01% and 0.1%), and is clearly problematic for variants seen in hundreds to thousands of individuals (0.1% to 1%).

Response: This concern is fully addressed; see our response in section *R3.h* above.

Comment R3.j. Finally, the text states that by removing variants “outside known regulatory elements”, PM1 becomes appropriate to use. However, this is an overly liberal usage of PM1, which states that a variant is in a “critical and well-established functional domain (e.g., active site of an enzyme) without benign variation”. The text describing PM1 in the ACMG guidelines paper further describes it as applying to “certain protein domains” that are “critical to protein function” and in which “all missense variants identified to date in these domains have been shown to be pathogenic”. It is debatable whether any regulatory element qualifies for use of this code, but even if there exist individual regulatory elements for which this code is appropriate, merely being inside “a regulatory element” would not be sufficient to invoke it, much as merely being inside “a protein domain” is also insufficient.

Response: We agree that asserting the functional impact of noncoding variants is highly challenging, especially among adjacent genomic positions within the same functional unit. This is exactly what our analysis suggested, i.e., current computational predictions are limited to identifying regional effects instead of positional effects. The restriction may be caused by the limited resolution of functional assays, which prevents the exact mapping of functional variants in a short genomic range. We have already mentioned this potential issue in the *Discussion* section of the previous manuscript.

Comment R3.k. In general, I believe this study falls well short of providing sufficient evidence that the DAVs as defined are reliably pathogenic. The concerns described above would be relevant to all claims of pathogenicity, but rigor is particularly important for variants that do not disrupt proteins and thus have intrinsically higher a priori probabilities of not being truly disease-causal. Such standards may be inconvenient (and clearly we all wish there was a large collection of truly highly penetrant non-coding variants), but cannot simply be ignored when drawing conclusions that depend on reliable definitions of pathogenicity.

Response: We have followed the standards in the field in developing the tools we have evaluated, who did not have access to data with any higher quality than us (or anyone else, as noted by R3 also). We believe that the patterns we have reported are robust and generalizable.

Reviewer #1 (Remarks to the Author):

The responses by the authors are very comprehensive and adequate. They have added a lot of additional evidence and comparisons into their analyses, including the use of GnomAD/ExAC data to set population frequency thresholds for variant classification. These results were in compliance with the ones retrieved when using the 1KG project's data, which shows the robustness of their method.

One of Reviewer #3's major concerns is that HGMD and ClinVar are not highly accurate resources and that they may contain too many false positives. However, the authors tested, first of all, the effect of using different allele frequency thresholds for the annotation of pathogenic variants into their analyses and did not notice any meaningful differences across the respective results. That indicates that there is not any apparent effect of high contamination of ClinVar or HGMD with false positives. Additionally, the existence of a certain ratio of false positives in a data source cannot be the concern or focus of the current manuscript.

Another major point of concern raised by Reviewer #3 has to do with associating high conservation degree of a non-coding single-base site with pathogenicity. In particular, the authors have mainly employed evolutionary constraints to characterise non-coding variants as more likely to be pathogenic. This approach has been justified on the base that two of the methods under study (CADD and LINSIGHT) have also used conservation data in their models. Moreover, previous work (Lindblad-Toh et. al., 2011) has shown that the probability of a highly conserved site across 29 mammals not to be under purifying selection is considerably low (<0.02) and even lower in their model (<0.006 , since they use conservation data across 58 species). Thus, their justification is scientifically sound. Overall, even though there's no trivial way of inferring the pathogenicity of non-coding variants, the authors follow the current standards in the field in order to provide additional insights in an area which is, to its most part, an uncharted territory.

Finally, with regards to complying with the ACMG guidelines the authors provide a satisfactory response. They mention firstly that there are no official guidelines from ACMG regarding the characterisation of pathogenic non-coding variants. Instead, they transfer the existing ACMG guidelines from coding variants into the non-coding ones. They explicitly state how they have employed the ACMG annotation into their annotation of non-coding variants so I find their approach clear and coherent.